RESEARCH

# Single-cell multiomics of the human retina reveals hierarchical transcription factor collaboration in mediating cell type-specific effects of genetic variants on gene regulation

Jun Wang[1,2†], Xuesen Cheng[1,2†], Qingnan Liang[1,2,3], Leah A. Owen[4], Jiaxiong Lu[1,2,3], Yiqiao Zheng[5], Meng Wang[1,2], Shiming Chen[5,6], Margaret M. DeAngelis[7], Yumei Li[1,2] and Rui Chen[1,2*]

†Jun Wang and Xuesen Cheng contributed equally.

*Correspondence:
ruichen@bcm.edu

[1] Human Genome Sequencing Center, Baylor College of Medicine, Houston, TX, USA
[2] Department of Molecular and Human Genetics, Baylor College of Medicine, Houston, TX, USA
[3] Verna and Marrs McLean Department of Biochemistry and Molecular Biology, Baylor College of Medicine, Houston, TX, USA
[4] Department of Ophthalmology and Visual Sciences, John A. Moran Eye Center, University of Utah, Salt Lake City, UT, USA
[5] Department of Ophthalmology and Visual Sciences, Washington University in St Louis, Saint Louis, MO, USA
[6] Department of Developmental Biology, Washington University in St Louis, Saint Louis, MO, USA
[7] Department of Ophthalmology, University at Buffalo the State University of New York, Buffalo, NY, USA

## Abstract

**Background:** Systematic characterization of how genetic variation modulates gene regulation in a cell type-specific context is essential for understanding complex traits. To address this question, we profile gene expression and chromatin accessibility in cells from healthy retinae of 20 human donors through single-cell multiomics and genomic sequencing.

**Results:** We map eQTL, caQTL, allelic-specific expression, and allelic-specific chromatin accessibility in major retinal cell types. By integrating these results, we identify and characterize regulatory elements and genetic variants effective on gene regulation in individual cell types. The majority of identified sc-eQTLs and sc-caQTLs display cell type-specific effects, while the cis-elements containing genetic variants with cell type-specific effects are often accessible in multiple cell types. Furthermore, the transcription factors whose binding sites are perturbed by genetic variants tend to have higher expression levels in the cell types where the variants exert their effects, compared to the cell types where the variants have no impact. We further validate our findings with high-throughput reporter assays. Lastly, we identify the enriched cell types, candidate causal variants and genes, and cell type-specific regulatory mechanism underlying GWAS loci.

**Conclusions:** Overall, genetic effects on gene regulation are highly context dependent. Our results suggest that cell type-dependent genetic effect is driven by precise modulation of both trans-factor expression and chromatin accessibility of cis-elements. Our findings indicate hierarchical collaboration among transcription factors plays a crucial role in mediating cell type-specific effects of genetic variants on gene regulation.

**Keywords:** Genetic variants, Gene regulation, Transcription factor collaboration, Cell type-specific effect, eQTL, caQTL, ASE, ASCA, Single-cell multiomics, The human retina

## Background

Gene regulation is cell type dependent [1], and the modulation of this process by genetic variation among individuals is a major contributor to complex traits and diseases [2–5]. Substantial progress has been made in mapping, annotation, and functional validation of regulatory variants [2, 6–9]. However, the mechanisms by which genetic variants modulate gene regulation in a cell type-specific context remain largely unclear [10, 11]. Indeed, prior in vivo studies conducted on bulk tissues have a limited ability to elucidate the cell type effects of gene regulation. This gap can be addressed by recent advances in single-cell omics technologies [1, 12–15]. Recent studies using single-cell omics technologies have generated cell atlases for different tissues and development stages, revealing regulatory elements in cell type/state resolution, facilitating the interpretation of noncoding variants [16–19]. Several pioneering studies have utilized single-cell sequencing to map expression quantitative trait loci (eQTL) or chromatin accessibility quantitative trait loci (caQTL) in individual cell type contexts, revealing the cell type/state-specific effects of genetic variants [20–26]. Despite these advancements, the mechanisms underlying cell type/state-specific effects of genetic variants remain elusive. To address these questions, we combined genomic sequencing with single-cell multiomics profiling of gene expression and chromatin accessibility to gain further insights (Fig. 1a).

Specifically, we conducted whole genome sequencing (WGS), single-nuclei RNA sequencing (snRNA-seq), and single-nuclei assay for transposase-accessible chromatin sequencing (snATAC-seq) on cells from the healthy retinae of 20 human donors. We mapped single-cell eQTLs (sc-eQTLs), single-cell caQTLs (sc-caQTLs), single-cell allelic-specific expression (sc-ASE), and single-cell allelic-specific chromatin accessibility (sc-ASCA) for major retinal cell types (Fig. 1a). Integration of these results enable us to genome-wide identify and characterize gene regulatory elements, as well as genetic variants affecting chromatin accessibility and gene expression within specific cell type contexts. The putative cis-regulatory elements were further validated through massively parallel reporter assays, and integrated with the analysis of ChIP-seq data of transcription factors (TFs) and histone modifications. Intriguingly, most of sc-QTLs identified are specific to one cell type, suggesting a significant proportion of variants modulate gene expression and chromatin accessibility in a cell type-dependent manner. Further analyses suggest that the cell type-specific effects of genetic variants may be attributed to cell type-specific modulation of transcription factor expression and chromatin accessibility of the affected cis-elements, through perturbations in the binding of TFs by the genetic variants. Lastly, by integrating the single-cell multiomics data, genetic association results, and Genome-Wide Association Studies (GWAS), we identified the enriched cell types, fine-mapped candidate causal variants and genes, and uncovered the regulatory mechanisms underlying GWAS loci.

## Results

### Single-nuclei multiomics profiling of 20 healthy human donor retinae

To profile gene expression and chromatin accessibility in a cell type-specific context, we performed snRNA-seq and snATAC-seq on the healthy retinae from 20 human donors (Fig. 1a, Additional file 2: Table S1). For snRNA-seq, upon quality control

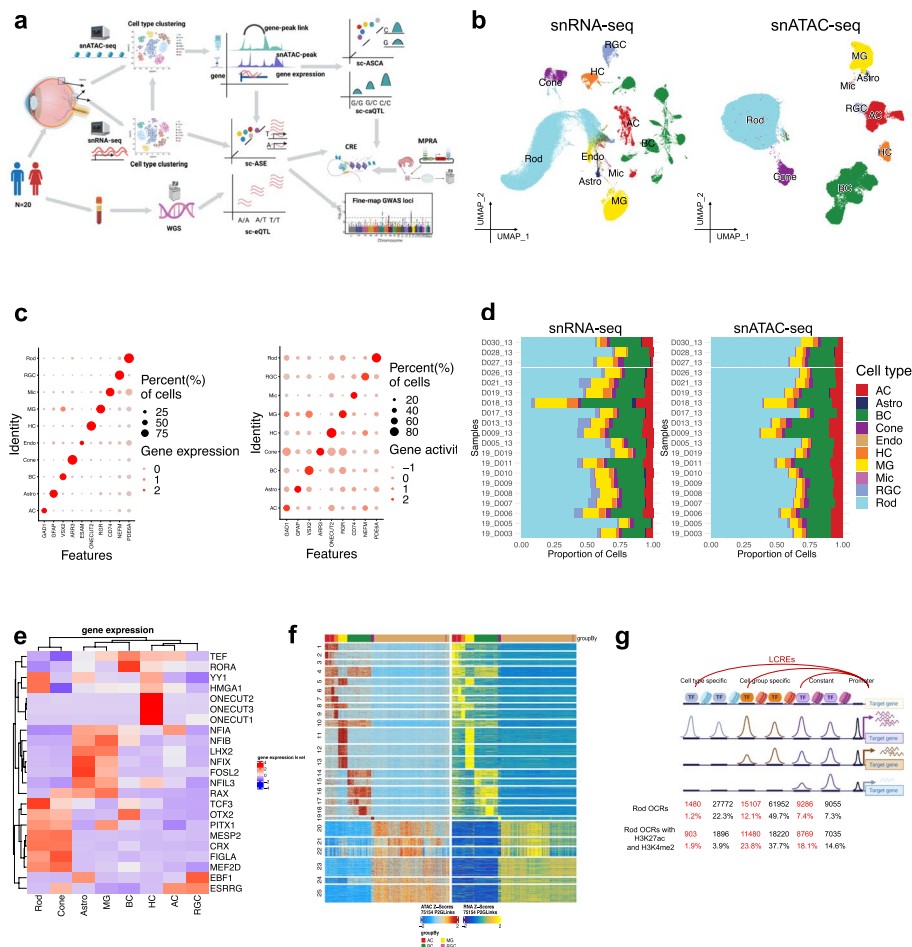

**Fig. 1** Profiling gene expression and chromatin accessibility of retinal cells. **a** Schematics of experiment design. **b** Uniform Manifold Approximation and Projection (UMAP) of cells from snRNA-seq and snATAC-seq. Each dot corresponds to a cell, colored by the annotated cell types. The same cell types from the two modalities are labeled with the same colors. $n = 192,792$ snRNA-seq cells. $n = 245,940$ snATAC-seq cells. **c** Dot plot showing the expression level and gene activity of marker genes for each major cell type, which were derived from snRNA-seq and snATAC-seq respectively. The dot color indicates the gene expression or activity level, while the dot size corresponds to the percentage of cells exhibiting that particular gene expression or activity. **d** Bar plot showing the distribution of cell type proportions across 20 donors based on snRNA-seq and snATAC-seq data respectively. Each cell type is represented by a distinct color. The number of cells per cell type per donor is listed in Additional file 2: Table S2. **e** Heatmap showing the gene expression level of the transcription factors across major retinal cell types. Those transcription factors were identified through chromVAR analysis and the correlation between motif enrichment and gene expression. **f** Heatmap showing the chromatin accessibility (left) and gene expression (right) of 75,154 significantly linked CRE-gene pairs across single-cell groups. Each row corresponds to a CRE-gene pair. Rows were clustered using *k*-means clustering ($k = 25$). Each column corresponds to a single-cell group. The color bar at the top of the heatmap indicates the cell type of a single-cell group. **g** The diagram showing proportions of Rod OCRs that are cell type-specific LCRE, cell type-specific non-LCRE, cell group-specific LCRE, cell group-specific non-LCRE, constant LCRE, and constant non-LCRE

(QC), a total of 192,792 nuclei were clustered into 10 major retinal cell classes, including rod photoreceptors (Rod), cone photoreceptors (Cone), bipolar cells (BC), amacrine cells (AC), horizontal cells (HC), müller glia cells (MG), retinal ganglion cells (RGC), astrocytes (Astro), endothelial cells, and microglia cells ("Methods",

Fig. 1b, Additional file 1: Fig. S1a). In parallel, snATAC-seq was performed for the same set of donor retinae. After QC, a total of 245,940 nuclei were clustered into 9 major retinal cell classes (Fig. 1b, Additional file 1: Fig. S1b). Consistent with the cell type annotation, canonical cell type marker genes show specific expression and gene activity in the corresponding cell clusters from snRNA-seq and snATAC-seq respectively [27] (Fig. 1c). Furthermore, the distributions of cell type proportions profiled by the two methods are highly concordant across the samples, ranging from 2.5% RGC to 55.2% Rod (Fig. 1d, Additional file 2: Table S2).

A total of 430,567 open chromatin regions (OCRs) were identified from the snATAC-seq data, ranging from 48,764 to 199,666 per cell type ("Methods", Additional file 2: Table S3). To assess the quality of these OCRs, we compared them with OCRs detected by bulk ATAC-seq [28]. The snATAC-seq OCRs exhibit high sensitivity, capturing most OCRs identified by bulk ATAC-seq (Additional file 1: Fig. S1c,d,e). Specifically, 74.9 and 84.2% of OCRs identified by bulk ATAC-seq of retina and macula tissues were detected in the snATAC-seq dataset respectively [28]. Additionally, 96.2% of putative active enhancers identified in previously published bulk seq data were found in the snATAC-seq OCR list [28] (Additional file 1: Fig. S1c). Consistent with Rod being the most abundant cell type in the retina, OCRs identified in Rod show the strongest correlation with those in the bulk retina data (a Pearson correlation of 0.69), while lower correlations were observed for the cell types that represent a small proportion of the total retinal cell population (for example, a Pearson correlation of 0.41 for RGC, Additional file 1: Fig. S1d). Conversely, 74.0% of the OCRs were only detected by snATAC-seq, indicating a large portion of OCRs are specific to a subset of cell types. Indeed, 51.5% of the snATAC-seq OCRs are unique to one cell type (Additional file 2: Table S3), which were largely missed by bulk ATAC-seq with a low detection rate of 14.3% (Additional file 1: Fig. S1e). To further evaluate the snATAC-seq OCRs, we investigated the enrichment of TF binding motifs within the OCRs for each cell type (Fig. 1e). Consistently, many identified TFs are previously shown to play cell type-specific role in the retina, such as OTX2, CRX, MEF2D in photoreceptor cells, ONECUT2 in HC, NFIA, NFIB, NFIX, and LHX2 in MG, supporting our findings [29–33].

Based on our multiomics dataset, we identified putative cis-regulatory elements (CREs) and their target genes by correlating OCR accessibility with nearby ($\pm 250$ kb) promoter accessibility or gene expression; we refer to these OCRs as linked CREs (LCREs) (Fig. 1f). Approximately 16.6% (71,274) of OCRs are linked to 13,405 target genes, averaging 5.9 LCREs per gene per cell type. As expected, LCREs are enriched for the CREs identified in previous studies, with 74.2 and 87.0% of the LCREs found in the ENCODE cCRE registry [6] and recent cCREs atlas [16] respectively (1.44- and 1.26-fold enrichment compared to all the OCRs, two-sided binomial test, $< 2.2 \times 10^{-16}$). Furthermore, LCREs are highly enriched with active enhancers. For example, 81.8% of LCREs in Rod carry the epigenetic modifications of active enhancers, concurrent H3K4me2 and H3K27ac, representing a 2.1-fold enrichment compared to all the OCRs (two-sided binomial test, $p < 2.2 \times 10^{-16}$, Fig. 1g). For each cell type, on average 5.9% of LCREs are in cell type-specific OCRs, 62.1% of LCREs are from OCRs shared by multiple cell types, and 32.0% of LCREs are located in constant OCRs (Fig. 1g). LCREs tend to be in more dynamic OCRs with overall 57.3% in differential accessible regions (DARs), representing

a 2.2-fold enrichment compared to all the OCRs ($p < 2.2 \times 10^{-16}$, Additional file 1: Fig. S1f).

### Significant proportion of sc-eQTLs are cell type-specific in retina

To profile genetic variation in the donors, WGS was conducted on each donor, yielding a total of 9.8 million genetic variants after QC (Additional file 1: Fig. S2a). To identify genetic variants that have an impact on gene expression, we mapped sc-eQTLs for each major retinal cell type. Due to the limited number of individuals available for our study, we focused on variants with allele frequency $\geq 0.1$ that are located within OCRs surrounding the genes ($\pm 250$ kb of gene transcription start site, TSS), totaling 421,004 variants, averaging 59.9 variants per gene and 2.8 variants per OCR per cell type.

A total of 14,377 sc-eQTLs that reach gene-level significance with false discovery rate (FDR) < 10% were identified. These sc-eQTLs were further grouped based on linkage disequilibrium (LD) ($r^2 > 0.5$) and their eGenes, resulting in 5688 independent sc-eQTL sets associated with 4069 sc-eGenes, ranging from 704 to 1175 sc-eQTL sets per cell type (Fig. 2a, b, Additional file 2: Table S4). The majority (86.1–91.8%) of sc-eGenes has only one sc-eQTL set per cell type (Additional file 1: Fig. S2b). Interestingly, most of sc-eQTLs are cell type-specific, with 87.0–92.3% identified in only one cell type (Fig. 2a). Furthermore, the remaining sc-eQTLs that are observed in multiple cell types are often shared among closely related cell types, such as between rod and cone photoreceptors (Additional file 1: Fig. S2c). Consistently, the effect of sc-eQTLs across cell types is correlated with the similarity between cell types (Fig. 2c), for example, a stronger correlation is observed between rod and cone photoreceptors (Pearson correlation $r = 0.6$). These results suggest that genetic variants have a more consistent impact on gene regulation among closely related cell types that share similar transcription programs. Interestingly, for sc-eQTLs located in non-promoter OCRs, those shared by multiple cell types tend to have greater effect sizes compared to those unique to one cell type (e.g., Rod, two-sided Wilcoxon rank sum test, $p = 1.88 \times 10^{-5}$, Fig. 2d, Additional file 2: Table S5). Consistently, for sc-eQTLs located in non-promoter OCRs, those shared by multiple cell types are located closer to the TSS of their eGenes than those unique to one cell type (e.g., Rod, two-sided Wilcoxon rank sum test, $p = 9.75 \times 10^{-11}$, Fig. 2e, Additional file 2: Table S6).

### Validation of sc-eQTLs with bulk eQTLs and sc-ASE

To evaluate the quality of sc-eQTLs, we compared them with the bulk eQTLs previously identified in retina and other tissues through bulk RNA-seq from the GTEx project [34]. sc-eQTLs are enriched for bulk eQTLs. On average, 35.6% of sc-eQTLs are overlapped with the bulk retina eQTLs (4.4-fold enrichment compared to background variants, two-sided binomial test $p < 1.2 \times 10^{-166}$) and 56.0% overlapped with the bulk eQTLs from all the 49 tissues (2.3-fold enrichment compared to background, two-sided binomial test $p < 2.1 \times 10^{-145}$, Fig. 2f). The proportion of overlap varies across cell types (Fig. 2f). As expected, the highest overlap (63.9%) is observed for the most abundant cell type, Rod, while the lowest overlap is observed for HC (49.0%) (Fig. 2f). Importantly, the effect directions of eQTLs are largely concordant across different cell types and tissues (Fig. 2g).

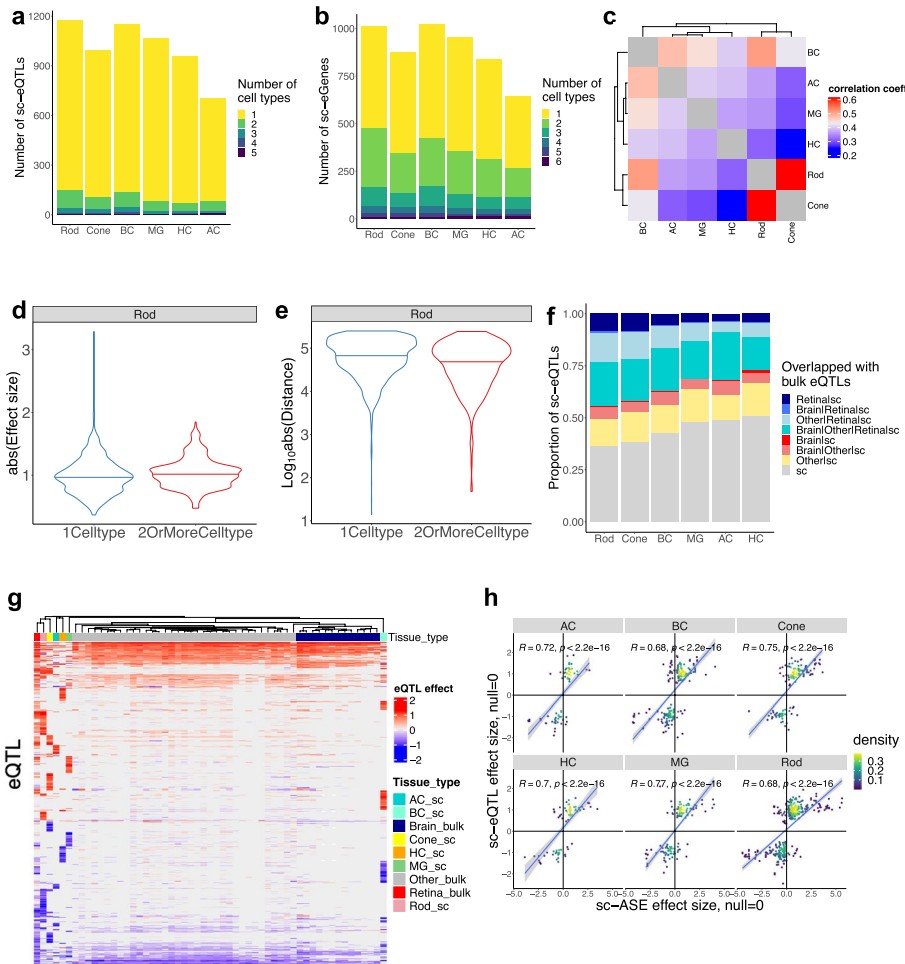

**Fig. 2** Identification of sc-eQTLs in retinal cell types. **a** Bar plot showing the number of independent index sc-eQTLs reaching gene-level FDR < 0.1 per cell type, colored by the number of cell types where a sc-eQTL is significant. **b** Bar plot showing the number of sc-eGenes reaching gene-level FDR < 0.1 per cell type, colored by the number of cell types where a sc-eGene is significant. **c** Heatmap showing the Pearson correlation of sc-eQTL effect size across retinal cell types. **d** Violin plot showing the distribution of absolute effect size of the sc-eQTLs located in non-promotor OCRs identified in Rod, colored by whether sc-eQTLs were identified in one ($n = 1969$) or more cell types ($n = 539$). To compare the effect size between the two types of sc-eQTLs, two-sided Wilcoxon rank sum test was performed, $p = 1.88 \times 10^{-5}$. **e** Violin plot showing the distribution of the distance between sc-eQTL and eGene TSS for the sc-eQTLs located in non-promotor OCRs identified in Rod, colored by whether sc-eQTLs were identified in one ($n = 1969$) or more cell types ($n = 539$). To compare the distance distribution between the two types of sc-eQTLs, two-sided Wilcoxon rank sum test was performed, $p = 9.75 \times 10^{-11}$. **f** Bar plot showing proportion of sc-eQTLs overlapping with bulk eQTLs, colored by the tissue types where the bulk eQTL is significant. sc: the identified sc-eQTLs. Other: other tissue. **g** Heatmap showing effect size of sc-eQTLs and the overlapped bulk eQTLs. Each row corresponds to an eQTL. Each column corresponds to a tissue or cell type. **h** Scatter plot showing the effect size of the overlapped sc-ASEs (*X* axis) and sc-eQTLs (*Y* axis) in the corresponding cell type. The Pearson correlation coefficient and *p*-value are indicated in the plot

We further validated these sc-eQTLs by comparing them with sc-ASEs. sc-eQTLs were found to be enriched for sc-ASEs. Among the sc-eQTLs tested for sc-ASEs, sc-ASEs were detected in 18.8–34.0% of the variants (with the highest overlap in Rod, 34.0%), on average 2.5-fold enrichment compared to the background variants (two-sided binomial test $p < 1.2 \times 10^{-12}$, Additional file 1: Fig. S2d). The effect size and direction are

positively correlated between the overlapped sc-eQTLs and sc-ASEs (Pearson correlation, $r$ in 0.68–0.77, $p < 2.2 \times 10^{-16}$), with most (82.5–94.2%) of the overlapped variants having the same effect direction (Fig. 2h, Additional file 1: Fig. S2d). Altogether, these results support that the majority of identified sc-eQTLs are likely to be true positives.

### Cell type-specific sc-eQTLs often reside in OCRs shared by multiple cell types

An interesting observation is that most (87.0–92.3%) of sc-eQTLs are unique to one cell type, while the associated sc-eGenes (94.6–98.9%) are almost always expressed in multiple cell types (Figs. 2a and 3a). Specifically, only a small proportion (1.8–6.0%) of sc-eQTLs and their associated sc-eGene expression exhibit the same pattern of cell type specificity. In over 90% of the cases, while the sc-eQTL is observed in one or a subset of cell types, the sc-eGenes are expressed in more cell types. Interestingly, different sc-eQTLs are often observed for the same sc-eGene in different cell types (36.4% of total sc-eQTLs) (Fig. 3b), and these sc-eQTLs tend to be located in different OCRs (34.0% of total sc-eQTLs, Additional file 1: Fig. S2e). This cannot be solely attributed to cell type-specific accessibility of the OCRs, as most of cell type-specific sc-eQTL variants reside in the OCRs accessible across multiple cell types. It is not primarily due to differential accessibility of the OCRs either, as only a small proportion (8.8–19.4%) of sc-eQTLs reside in the DARs specific to the corresponding cell types. In fact, merely a small fraction (11.4%) of sc-eQTLs reside in OCRs with matching cell type specificity as those of the sc-eQTLs (Fig. 3c). For example, the variant rs10793810 is identified as a sc-eQTL specific to MG, influencing the expression of *SLC27A6*. This variant was predicted to enhance the binding of *FOXP2* (highly expressed in MG), to a MG-specific LCRE of *SLC27A6* (Fig. 3d). In contrast, the majority (89.1%) of sc-eQTLs are located within OCRs shared among multiple cell types (Fig. 3c), suggesting that the modulation of gene expression by genetic variants is driven by cellular activity of trans-factors and accessibility of cis-elements. For example, the variant rs62308155 is identified as a sc-eQTL specific to Rod, affecting

(See figure on next page.)
**Fig. 3** Cell type-specific sc-eQTLs are often within OCRs shared by multiple cell types. **a** Bar plot showing proportion of sc-eQTLs associated with sc-eQTLs significant in one or more cell types, colored by the number of cell types where the sc-eGene is expressed. **b** Bar plot showing proportion of sc-eQTLs associated with sc-eGenes significant in one or more cell types, colored by the number of cell types where the sc-eQTL is significant. **c** Bar plot showing proportion of sc-eQTLs associated with sc-eQTLs significant in one or more cell types, colored by the number of cell types where the OCR containing the sc-eQTL is accessible. **d** Box plot showing the expression of *SLC27A6* in 20 individuals with different genotype of rs10793810 in MG. FOXP2 motif position weight matrix aligned to reference and alternative allele at rs10793810. Violin plot showing the expression level of *SLC27A6* and *FOXP2* across retinal cell types ($n = 50{,}000$ cells). Genome track of *SLC27A6* locus showing cell type-specific chromatin accessibility and positioning of rs10793810 within an OCR that is a predicted LCRE of *SLC27A6*. **e** Box plot showing the expression of *REST* in 20 individuals with different genotype of rs62308155 in Rod and Cone. NR3C1 motif position weight matrix aligned to reference and alternative allele at rs62308155. Violin plot showing the expression level of *REST* and *NR3C1* across retinal cell types ($n = 50{,}000$ cells). Genome track of *REST* locus showing cell type-specific chromatin accessibility and positioning of rs62308155 within an OCR that is a predicted LCRE of *REST*. **f** Violin plot showing distribution of gene expression levels of TFs in Rod and another cell type. These TFs have binding motifs perturbed by a sc-eQTL that has effect specifically in Rod but not in the other cell type. One-sided Wilcoxon rank sum test was performed to detect difference of gene expression of these TFs between Rod and the other cell type. The *p*-value and sample size *n* are indicated in the figure

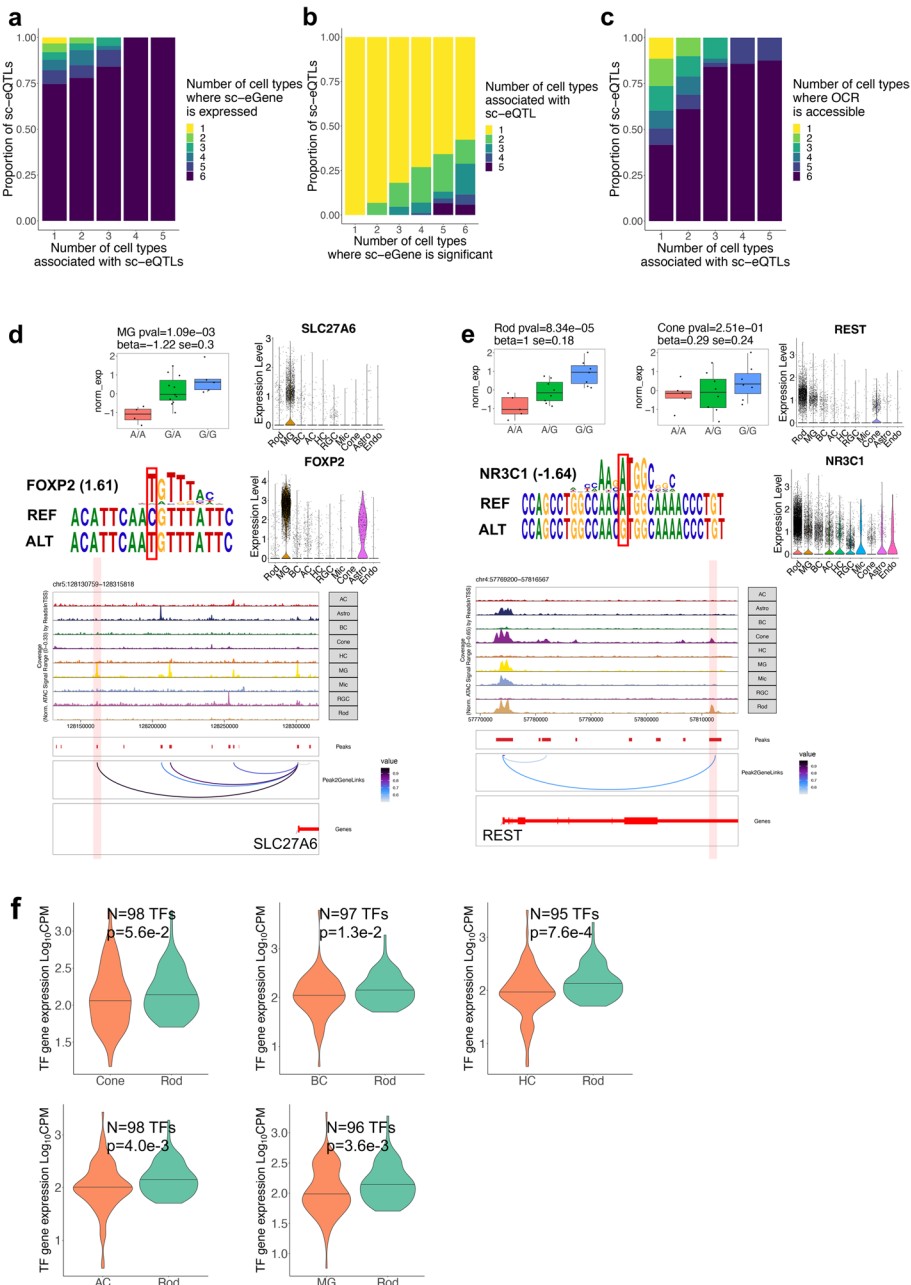

**Fig. 3** (See legend on previous page.)

the expression of *REST*. This variant likely disrupts the binding of *NR3C1*, which is highly expressed in Rod but minimally in Cone, to a LCRE accessible in both Rod and Cone (Fig. 3e). Furthermore, genome-widely, TFs with motifs perturbed by genetic variants tend to have higher expression levels in the cell types where the variants have sc-eQTL effects compared to the cell type where the variants do not have an effect, even though the OCRs containing the sc-eQTLs are accessible in both cell types (e.g., Rod, one-sided Wilcoxon rank sum test, $p < 0.057$, Fig. 3f, Additional file 2: Table S7).

## Significant proportion of sc-caQTLs are cell type-specific in retina

In parallel with sc-eQTL analysis, we also conducted sc-caQTL analysis to iden-tify genetic variants that affect chromatin accessibility, by examining the association between each OCR and common variants within it for each major retinal cell type. A total of 174,419 OCRs (ranging from 54,716 to 95,020 OCRs per cell type) and the same set of variants tested for sc-eQTLs were analyzed ("Methods"). Upon genome-wide multiple testing corrections, 23,287 sc-caQTLs were identified (FDR < 10%). These sc-caQTLs were grouped into 12,482 independent sc-caQTL sets mapped in 10,298 OCRs based on LD ($r^2 > 0.5$), ranging from 391 to 4789 sc-caQTLs sets per cell type (Fig. 4a, b and Additional file 2: Table S8). The majority (88.0%) of caQTL-associated peaks (sc-caPeaks), which are OCRs containing caQTLs in this study, are associated with only

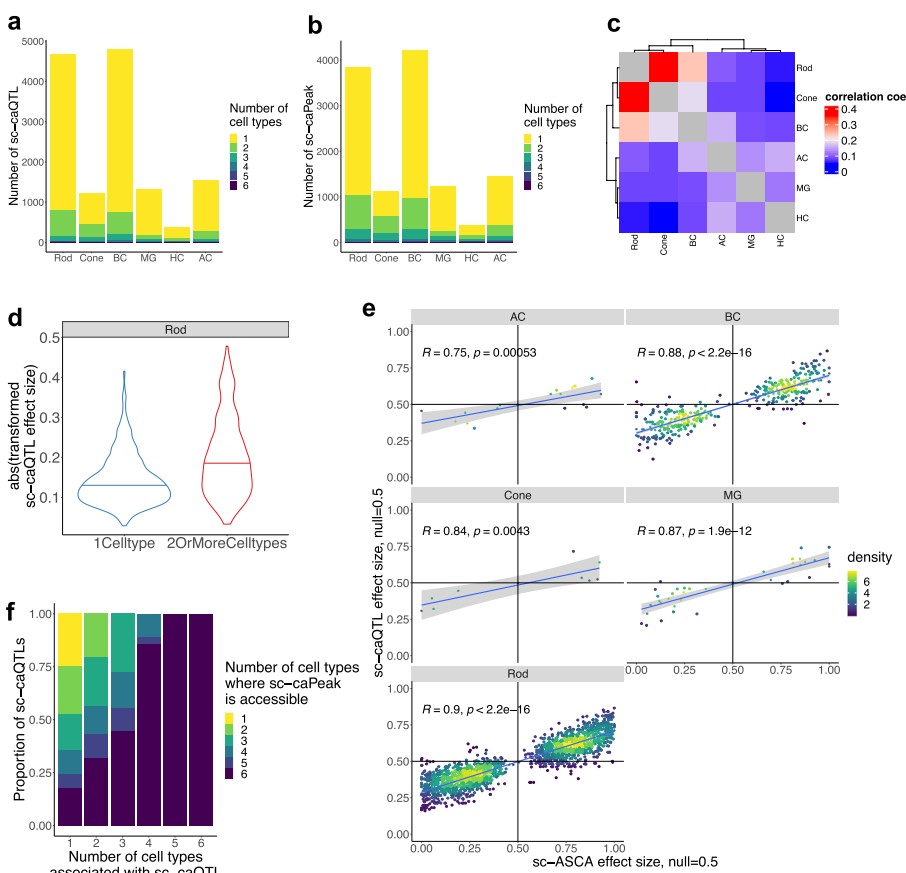

**Fig. 4** Identification of sc-caQTL in retinal cell types. **a** Bar plot showing the number of independent index sc-caQTLs reaching genome-level FDR < 0.1 per cell type, colored by the number of cell types where a sc-caQTL is significant. **b** Bar plot showing the number of sc-caPeaks reaching genome-level FDR < 0.1 per cell type, colored by the number of cell types where a sc-caPeak is significant. **c** Heatmap showing the Pearson correlation of sc-caQTL effect size across retinal cell types. **d** Violin plot showing the distribution of absolute effect size of the sc-caQTLs located in non-promotor OCRs identified in rod cells, colored by whether sc-caQTLs were identified in one ($n = 6102$) or more cell types ($n = 2085$). To compare the effect size of the two types of sc-caQTLs, two-sided Wilcoxon rank sum test was performed, $p = 1.09 \times 10^{-132}$. **e** Scatter plot showing the effect size of sc-ASCAs (*X* axis) and the population effect size of the overlapped sc-caQTLs (*Y* axis). The Pearson correlation coefficient and *p*-value are indicated in the figure. **f** Bar plot showing proportion of sc-caQTLs associated with sc-caQTLs that are significant in one or more cell types, colored by the number of cell types where the caPeak is accessible

one sc-caQTL set (Additional file 1: Fig. S3a). Similar to sc-eQTLs, the majority of sc-caQTLs are cell type-specific with 62.3–85.7% being unique to one cell type. The effect sizes of sc-caQTLs are correlated across cell types, with stronger correlation observed between more closely related cell types (Fig. 4c and Additional file 1: Fig. S3b). Compared to the correlation of sc-eQTL effect sizes across cell types, those of sc-caQTLs appear to be smaller. This suggests that sc-caQTL effects show higher cell type specificity than sc-eQTLs, possibly due to the greater variation in chromatin accessibility across cell types compared to gene expression. Similar to sc-eQTLs, for sc-caQTLs located in non-promoter OCRs, those shared among multiple cell types displayed significantly greater effect than those unique to one cell type (e.g., Rod, two-sided Wilcoxon rank sum test, $p = 1.09 \times 10^{-132}$, Fig. 4d, Additional file 2: Table S9).

### Validation of sc-caQTLs with sc-ASCA

To assess the quality of the identified sc-caQTLs, we compared them with sc-ASCAs. sc-ASCAs are detected in 8.7–41.8% of the sc-caQTLs tested for sc-ASCAs (with the highest overlapping rate in Rod, 41.8%), on average 15.9-fold enrichment compared to the background variants (two-sided binomial test, $p < 6.8 \times 10^{-11}$, Additional file 1: Fig. S3c). The effect size and direction of sc-ASCAs are positively correlated with those of the overlapped sc-caQTLs (Pearson correlation, $r$ in 0.75–0.90, $p < 5.4 \times 10^{-4}$), with the majority (82.4–100%) of the overlapped variants showing the same effect direction (Fig. 4e, Additional file 1: Fig. S3c). These findings support that identified sc-caQTLs are indeed enriched of variants associated with change in chromatin accessibility. Conversely, 33.3–54.5% of the identified sc-ASCAs overlap with sc-caQTLs, depending on cell type. Interestingly, OCRs containing sc-ASCA-only variants (not overlapping with sc-caQTL) were significantly wider than those containing variants that are both sc-ASCA and sc-caQTL (Additional file 1: Fig. S3d, e.g., one-sided Wilcoxon rank sum test, $p = 7.7 \times 10^{-13}$ in Rod). This observation suggests that variants in wider OCRs tend to have a local effect, while variants in narrow OCRs are more likely to affect accessibility of the entire OCRs.

### Cell type-specific sc-caQTLs can reside in OCRs accessible in multiple cell types

Interestingly, similar to sc-eQTLs, most (62.3–85.7%) of sc-caQTLs are unique to one cell type, while the majority (74.8%) of sc-caPeaks are accessible in multiple cell types (Fig. 4a, f). Specifically, only 24.4% of sc-caQTLs and their caPeak accessibility share the same pattern of cell type specificity. The remaining 75.6% of sc-caQTLs were found in one or a subset of cell types, while their caPeaks are accessible in more cell types (Fig. 4f). The cell type-specific sc-caQTLs cannot be solely attributed to the differential accessibility of OCRs either, since a small proportion (14.9–34.3%) of sc-caQTLs were observed in the DARs of the corresponding cell types. Interestingly, for sc-caPeaks shared across multiple cell types, a higher proportion is associated with common sc-caQTL variants, compared to those associated with cell type-specific sc-caQTL variants (23.4% vs 10.4% of sc-caPeak per cell type, one-sided Fisher's exact test, $p = 2.48 \times 10^{-17}$). As an example where cell type specificity of sc-caQTLs matches with that of their residing OCRs, the variant rs12447029 exhibits a sc-caQTL effect specifically in MG through strengthening the binding of NFE2L2 (which is highly expressed in multiple cell types)

to its residing OCR, which is specifically accessible in MG (Fig. 5a). Consistently, the corresponding OCR is a LCRE of *GRIN2A*, and rs12447029 is a sc-eQTL for *GRIN2A* in MG (Additional file 1: Fig. S3e). On the other hand, the cell type specificity of many sc-caQTLs cannot be solely attributed to the cell type specificity of the residing OCRs. A large proportion (68.3%) of sc-caQTLs are unique to one cell type but reside in the

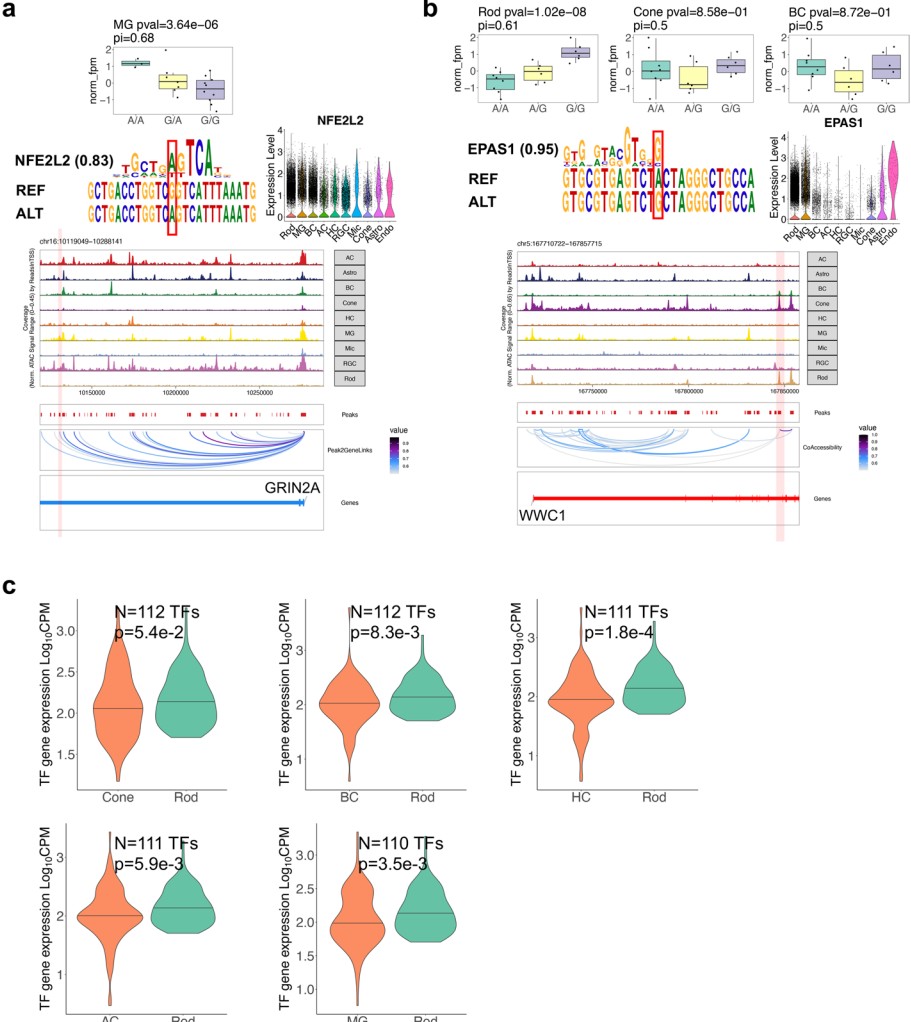

**Fig. 5** Cell type-specific effect of sc-caQTLs can be driven by trans-factors. **a** Box plot showing the chromatin accessibility of the OCR harboring rs12447029 in 20 individuals with different genotype of rs12447029 in MG. NFE2L2 motif position weight matrix aligned to reference and alternative allele at rs12447029. Violin plot showing the expression level of *NFE2L2* across retinal cell types. Genome track of *GRIN2A* locus showing cell type-specific chromatin accessibility and positioning of rs12447029 within an OCR which is a predicted LCRE of *GRIN2A*. **b** Box plot showing the chromatin accessibility of the OCR harboring rs6859300 in 20 individuals with different genotype of rs6859300 in Rod, Cone, and BC. EPAS1 motif position weight matrix aligned to reference and alternative allele at rs6859300. Violin plot showing the expression level of *EPAS1* across retinal cell types. Genome track of *WWC1* locus showing cell type-specific chromatin accessibility and positioning of rs6859300 within an OCR which is a predicted LCRE of *WWC1*. **c** Violin plot showing distribution of gene expression levels of TFs in Rod and another cell type. These TFs have binding motifs perturbed by a sc-caQTL that has effect specifically in Rod but not in the other cell type. One-sided Wilcoxon rank sum test was performed to detect difference of gene expression levels of these TFs between Rod and the other cell type. The *p*-value and sample size *n* are indicated in the figure

OCRs accessible in multiple cell types, indicating the modulation of chromatin accessibility by genetic variants is often cell type-dependent and likely through perturbing the binding of cell type-specific trans-factors (Fig. 4f). For example, the variant rs6859300 affects the chromatin accessibility of its residing OCR specifically in Rod, despite this OCR being accessible in multiple cell types (i.e., Rod, Cone, and BC), potentially through enhancing the binding of EPAS1, which is highly expressed in Rod but lowly expressed in Cone and BC (Fig. 5b). Consistently, the corresponding OCR is a LCRE of *WWC1*, and rs6859300 is a sc-eQTL of *WWC1* in Rod (Additional file 1: Fig. S3f). Furthermore, TFs with motifs perturbed by genetic variants tend to have higher expression levels in the cell types where the variants exhibit sc-caQTL effects, compared to the cell type where the variants do not have an effect, even though the OCRs containing the sc-caQTLs are accessible in both cell types. This result supports the role of trans-factors in driving cell type-specific sc-caQTL effects genome-widely (e.g., Rod, one-sided Wilcoxon rank sum test, $p < 0.054$, Fig. 5c, Additional file 2: Table S10).

In addition, to address potential bias in sc-caQTL detection introduced by cell type abundance difference, we applied a random down-sampling approach to repeat sc-caQTL analysis. The results obtained through this down-sampling analysis consistently aligned with our above findings based on the original dataset, confirming the robustness of the observed sc-caQTL patterns (see "Assessing the potential bias in caQTL detection introduced by cell type abundance differences" section in Additional file 1: Supplementary Note, Additional file 1: Fig. S4).

### Interaction among OCRs

Previous studies suggested that multiple regulatory elements can be regulated by a single genetic variant [11]. One possible mechanism is that the accessibility of a "master" element influences the accessibility of neighboring "dependent" elements [11]. To investigate this phenomenon in our dataset, we identified 2511 dependent regions associated with 1942 master regions ("Methods", Additional file 2: Table S11). Among these, 360 master regions that are LCREs are associated with 427 dependent regions that are also LCREs of the same genes. Notably, we observed a significant enrichment of sc-caQTLs associated with the dependent OCRs (e.g., 1.8-fold enrichment compared to the background variants in Rod, two-sided binomial test, $p = 1.48 \times 10^{-88}$) as well as dependent LCREs (e.g., 1.7-fold enrichment compared to background in Rod, two-sided binomial test, $p = 4.49 \times 10^{-12}$), suggesting the association between sc-caQTL/master elements and dependent elements is not random (Fig. 6a). Furthermore, a significant proportion (59.7%) of the master OCRs are co-accessible with at least one of the corresponding dependent OCRs (correlation $\geq 0.2$, FDR $< 0.05$), showing a 2.8-fold enrichment compared to the background (the co-accessibility of two random OCRs within a 250-kb region, two-sided binomial test, $p < 2.2 \times 10^{-16}$), further supporting the interactions between master and dependent OCRs. The effect size of sc-caQTLs on the master regions and dependent regions are positively correlated (an average correlation coefficient of 0.60, $p = 1.5 \times 10^{-7}$), and the majority (65.0–82.0%) of sc-caQTLs having the same effect direction on the master and dependent regions (Fig. 6b). Furthermore, we observed a slightly higher enrichment in DARs in the master regions compared to the dependent regions (one-sided Fisher's exact test, caPeak: $p = 7.1 \times 10^{-3}$; caPeaks

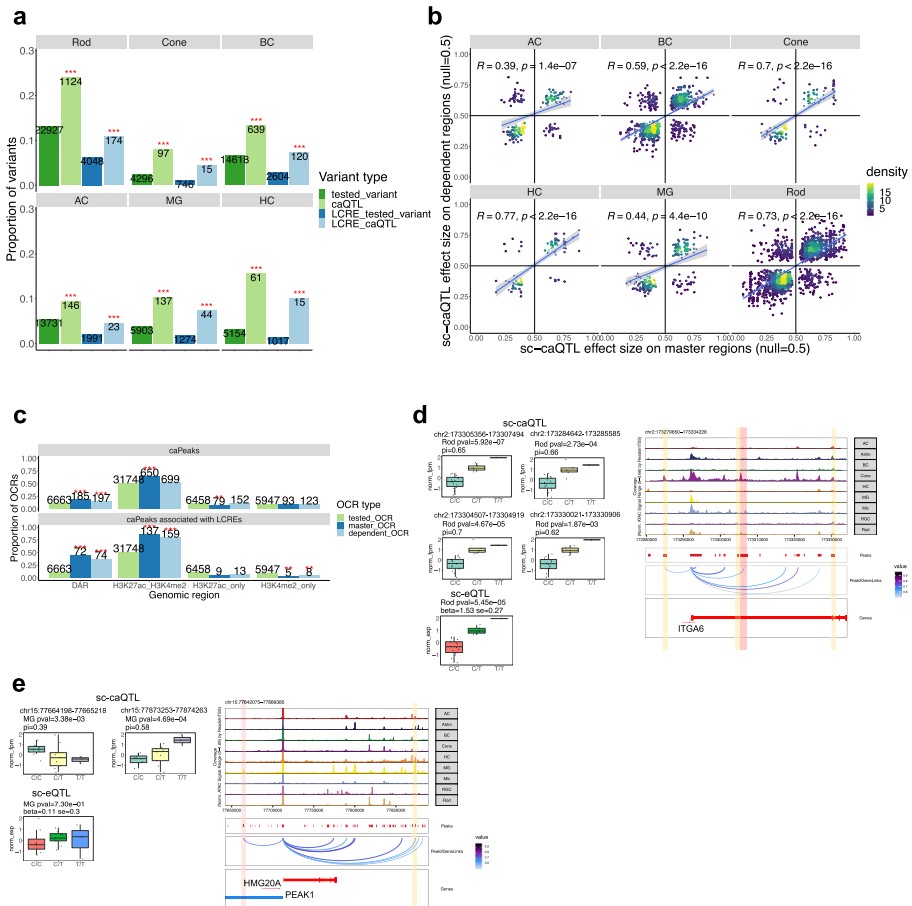

**Fig. 6** The sc-caQTLs can have effects on multiple genomic regions. **a** Bar plot showing the proportion of sc-caQTLs or the background variants affecting dependent OCRs and the proportion of sc-caQTLs or the background variants affecting dependent LCREs. The numbers of variants are indicated in the plot. Two-sided binomial test was performed to compare sc-caQTL with the background variants. "*": $0.05 > p \geq 0.01$, "**": $0.01 > p \geq 0.001$, "***": $p < 0.001$. **b** Scatter plot showing the effect size of sc-caQTLs on master regions (*X* axis) and on dependent regions (*Y* axis). **c** Bar plot showing the proportion of the master or dependent OCRs that are DARs or have concurrent H3K27ac and H3K4me2 modifications in Rod. The numbers of caPeaks with different features are indicated in the plot. Two-sided binomial test was performed to compare the proportion of master_OCR or dependent_OCR with the proportion of tested_OCR. **d** Box plot showing the chromatin accessibility of one master OCR and three dependent OCRs in 20 individuals with different genotype of rs7596259 in Rod. Box plot showing the gene expression level of *ITGA6* in 20 individuals with different genotype of rs7596259. Genome track of *ITGA6* locus showing cell type-specific chromatin accessibility of the master OCR (red) and three dependent OCRs (yellow). The master OCR and one of the dependent OCRs are the predicted LCREs of *ITGA6*. **e** Box plot showing the chromatin accessibility of one master OCR and one dependent OCR in 20 individuals with different genotype of rs1493699 in Rod. Box plot showing the gene expression level of *PEAK1* in 20 individuals with different genotype of rs1493699. Genome track of *PEAK1* locus showing cell type-specific chromatin accessibility of the master OCR (red) and one dependent OCR (yellow). The master OCR and the dependent OCR are the predicted LCREs of *PEAK1*

associated with LCREs: $p = 0.082$), as well as an enrichment in active enhancer epigenetic modifications (the concurrent H3K27ac and H3K4me2, one-sided Fisher's exact test, caPeak: $p = 1.48 \times 10^{-10}$; caPeaks associated with LCREs: $p = 0.10$) (Fig. 6c).

While the majority (66.5–87.7%) of master regions are associated with one dependent region, some master regions have multiple dependent regions. For example, the sc-caQTL variant rs7596259 increases the accessibility of its residing master region and is

also associated with the increased accessibility of three other dependent regions in Rod (Fig. 6d). This sc-caQTL variant is also a sc-eQTL, leading to increased gene expression of *ITGA6* in Rod, suggesting some of the affected regions may play a role in regulating gene expression (Fig. 6d). Indeed, the master region (chr2:173305356–173307494) and one of the dependent regions (chr2:173284642–173285585) are predicted LCREs of *ITGA6* (Fig. 6d). Interestingly, sc-caQTLs that affect multiple regions in the same effect direction are more likely to overlap with sc-eQTLs in the corresponding cell type compared to sc-caQTLs that affect multiple regions in different effect directions (in Rod 15.9% vs. 4.2%, two-sided binomial test $p = 2.25 \times 10^{-8}$). This observation suggests that compensatory effects between cis-elements associated with opposite effect directions may neutralize their impact on gene expression. For example, the sc-caQTL variant rs1493699 reduces the accessibility of its residing master region (chr15:77664198–77665218) and is associated with increased accessibility of a dependent region (chr15:77873253–77874263) in MG (Fig. 6e). Although both elements are LCREs of *PEAK1*, this sc-caQTL is not a sc-eQTL of *PEAK1*, suggesting that the two LCREs might compensate for each other, resulting in no overall change in gene expression (Fig. 6e). In addition, the affected OCRs of the majority (71.4%) of sc-caQTLs that impact multiple OCRs and also overlap with cs-eQTLs exhibit co-accessibility, presenting a 1.23-fold enrichment compared to those of sc-caQTLs that impact multiple peaks but do not overlap with cs-eQTLs (two-sided binomial test, $p = 1.96 \times 10^{-5}$). This result provides further evidence supporting the idea that multiple OCRs may interact jointly to determine a cs-eQTL.

### Massively parallel reporter assays of OCRs harboring sc-eQTLs

To validate the results of our association studies, we conducted Massively Parallel Reporter Assays (MPRAs) to assess the regulatory activities of 931 OCRs housing the index sc-eQTLs identified in human rod cells [35]. These sequences were tested in explanted mouse retinas during postnatal day 0 to 8, which primarily consist of rod cells (Fig. 7a). The MPRA library was designed with synthesized 224 bp oligonucleotides (oligos) centered on the peak summit of these OCRs (based on the human hg19 reference genome), along with positive and negative control sequences ("Methods"). Following the MPRAs, we calculated an activity score for each library sequence and normalized it based on the activity of the basal Crx promoter ("Methods"). As a result, we identified 258 enhancers and 66 silencers that showed at least a twofold higher or lower activity than the basal activity (i.e., activity of Crx basal promoter, $q$-value < 0.05), as well as 607 inactive sequences that exhibited activity within a twofold change of the basal activity or not significant different from the basal activity ($q$-value ≥ 0.05) (Fig. 7b, c, Additional file 2: Table S12). A high fraction of OCR sequences identified as inactive could be attributed to several factors: (1) limited sensitivity of the experimental system; (2) differential regulatory activity of CREs between human and mouse; and (3) false positives in identified sc-eQTLs due to a small cohort.

By integrating the MPRA result with bulk ChIP-seq data of the adult human retina, we observed that the validated enhancers are significantly enriched of the binding of photoreceptor-specific TFs, such as OTX2 (68.2% vs. 41.4%, one-sided Fisher's exact test, $p = 2.7 \times 10^{-13}$), CRX (35.7% vs. 15.7%, one-sided Fisher's exact test, $p = 1.8 \times 10^{-10}$) and MEF2D (28.7% vs. 12.7%, one-sided Fisher's exact

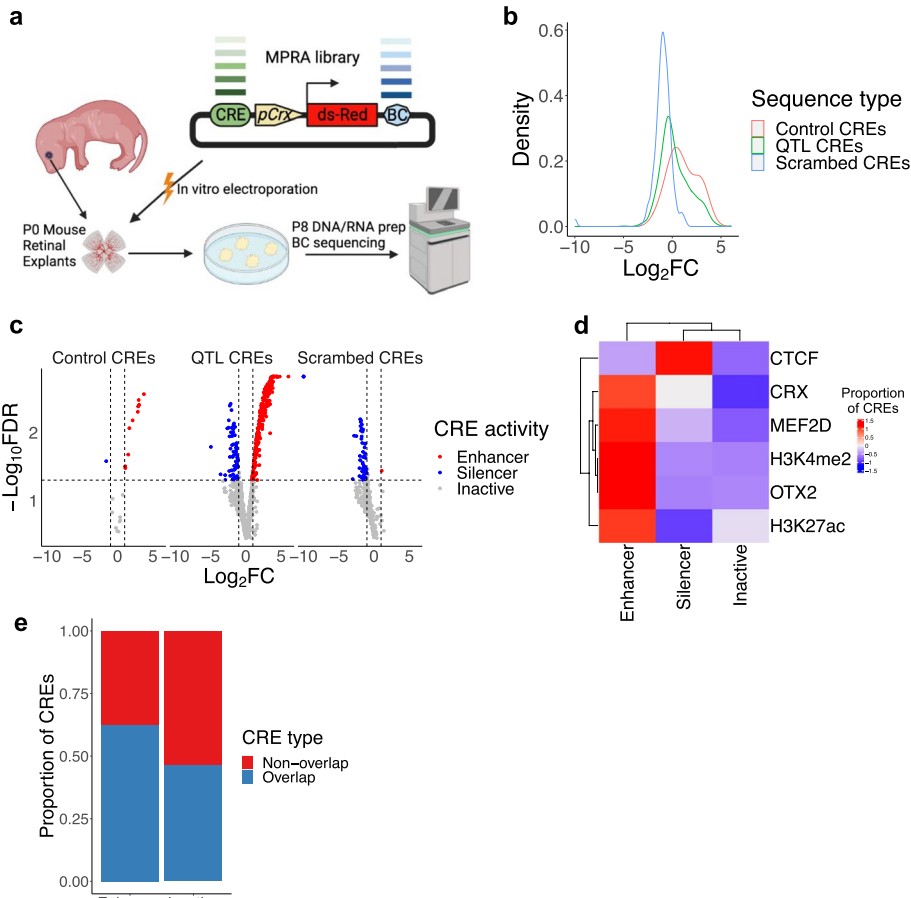

**Fig. 7** Massive parallel reporter assays (MPRAs) of cis-regulatory elements. **a** A schematic plot showing the workflow of MPRAs. **b** Density plot showing the distribution of $log_2FC$ values of the 1251 tested sequences, colored by the sequence type. **c** Volcano plot showing the $log_2FC$ value (X axis) and the $-log_{10}FDR$ value (Y axis) of each tested sequence ($n = 1251$). Each dot corresponds to a tested sequence, colored by the activity of the sequence. **d** Heatmap showing the scaled proportion of the regulatory sequences (i.e., Enhancer, Silencer, and Inactive) overlapped with different transcription factor bindings (i.e., CTCF, CRX, MEF2D, and OTX2) and histone modifications (i.e., H3K4me2 and H3K27ac). **e** Bar plot showing the proportion of the sequences with the co-binding of OTX2 and at least one of cell type/context-specific factors, CRX, NEF2D, and NRL (Overlap) and the remaining sequences (Non-overlap)

test, $p = 3.3 \times 10^{-8}$), compared to the inactive sequences (Fig. 7d). Additionally, a higher proportion of these enhancers exhibit active enhancer epigenetic markers, namely concurrent H3K27ac and H3K4me2, than the inactive sequences (76.4% vs. 59.5%, one-sided Fisher's exact test, $p = 9.7 \times 10^{-7}$, Fig. 7d). In contrast, the validated silencers are enriched of CTCF binding, compared to the inactive sequences (one-sided binomial test, 71.2% vs. 55.5%, one-sided Fisher's exact test, $p = 9.4 \times 10^{-3}$, Fig. 7d). These results provide experimental evidence to support that a significant proportion of OCRs harboring sc-eQTLs are indeed active CREs, suggesting that perturbation of these elements by genetic variants may lead to changes in gene expression. Interestingly, the validated enhancers are enriched of the co-binding of an early lineage-determining factor, OTX2, with at least one cell type/context-specific trans-factors, i.e., CRX, MEF2D, and NRL, compared to the inactive elements (62.5% vs. 46.2%, one-sided Fisher's exact

test, $p = 6.2 \times 10^{-4}$, Fig. 7e). This result indicates the potential role of OTX2 as a pioneer factor. These findings also suggest that collaboration between pioneer factors/early lineage-determining factors and cell type/context-specific factors is crucial for gene regulation.

**Prioritizing causal variants and cell type context underlying GWAS loci**

Our single-cell multiomics dataset provides unique opportunities to fine-map GWAS loci in a cell type-specific context. We first investigated the enrichment of GWAS loci associated with 11 eye traits or disorders in specific cell types [36–42], utilizing chromatin accessibility and gene expression of individual cell types respectively [43–47] ("Methods"). Interestingly, the analysis of chromatin accessibility and gene expression revealed consistent cell type enrichment patterns of GWAS loci (Fig. 8a, b, and Additional file 1: Fig. S5a, *q*-value < 0.1). Specifically, traits related to primary open-angle glaucoma (POAG), such as cup areas (CA) and vertical cup-disc ratio (VCDR) of optic nerve, intraocular pressure (IOP), and POAG itself, displayed enrichment in DARs, OCRs, and/or genes expression of astrocytes and MG ($p < 9.7 \times 10^{-3}$, *q*-value < 0.1, Fig. 8a, b). Loci associated with refractive error and myopia [42] exhibited enrichment in DARs, OCRs, and/or genes expression of most major retinal cell types ($p < 8.2 \times 10^{-3}$, *q*-value < 0.1) (Fig. 8a, b and Additional file 1: Fig. S5a). Loci related to choroid/retina disorders, retinal detachments/breaks, and retinal problems [41], showed enrichment in DARs of MG (Fig. 8a, $p < 7.2 \times 10^{-3}$, *q*-value < 0.1).

To identify causal candidate variants and target genes of GWAS loci within a cell type-specific context, we fine-mapped GWAS variants associated with three eye diseases, glaucoma [36], age-related macular degeneration [40], and refraction error/myopia [42]. We incorporated functional annotation (including OCR and LCRE derived from single-cell multiomics data) of variants to nominate potential GWAS causal variants [48]. As a result, 816 variants with posterior inclusion probability (PIP) > 0.1 were identified, which contain potential causal variants and prioritize the variants in regulatory regions (Additional file 1: Fig. S5b,c,d). Among them, 38 variants located in 28 LD blocks overlapped with sc-caQTL, sc-eQTL, and/or sc-ASCA (Additional file 1: Supplementary Note). From the 38 variants, we further focused on 28 variants that have the highest PIP values within their respective LD blocks (Fig. 8c, Additional file 1: Fig. S5e,f, Additional file 2: Table S13). Twenty-two (78.6%) of these 28 variants are located within regions displaying epigenetic modifications, H3K27ac and/or H3K4me2, supporting their regulatory roles (Additional file 2: Table S13). To identify target gene candidates, 20 of the 28 variants were linked to 24 target genes through sc-eQTLs, LCREs, and gene annotation (Additional file 2: Table S14). As expected, 16 (66.7%) of the 24 candidate genes are the nearest genes adjacent to the variants. Furthermore, 6 of these 24 genes are also supported by the colocalization of their retinal bulk eQTL signals with GWAS signals (Additional file 2: Table S14). For example, the variant rs511217 is a fine-mapped variant associated with refraction error and myopia (PIP = 0.22). This variant is a sc-eQTL of *KCNA4* and a nominal significant sc-caQTL of its residing OCR in BC. The corresponding OCR is a predicted LCRE of *KCNA4*. Consistently, the GWAS signal is colocalized with the retinal bulk eQTL signal of *KCNA4* (Additional file 1: Fig. S6).

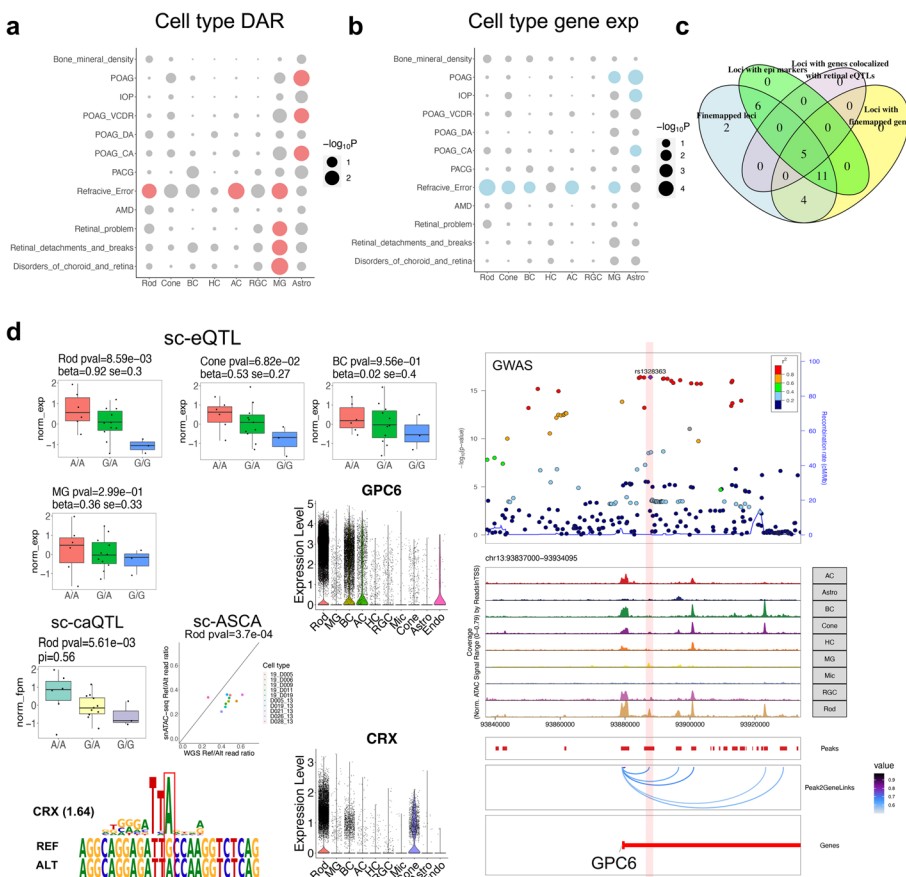

**Fig. 8** Cell type enrichment and causal variant prioritization underlying GWAS loci. **a** Dot plot showing the enrichment of 11 eye-related and one control GWAS loci in the DARs of retinal cell types, identified by LDSC. POAG: primary open-angle glaucoma. IOP: intraocular pressure. VCDR: vertical cup-disc ratio of optic nerve. CA: cup area of optic nerve. DA: disc area of optic nerve. PCAG: primary angle closure glaucoma. AMD: age-related macular degeneration. The cell types enriched of GWAS traits with *q*-value < 0.1 are highlighted in red. **b** Dot plot showing the enrichment of 11 eye-related and one control GWAS loci, based on cell type specificity of gene expression derived from snRNA-seq data with MAGMA.Celltyping. The cell types enriched of GWAS traits with *q*-value < 0.1 are highlighted in blue. **c** Venn diagram showing the features of the fine-mapped GWAS loci that are overlapped with sc-QTL and/or sc-ASCA. **d** Box plot showing the gene expression level of *GPC6* in 20 individuals with different genotype of rs1328363 in Rod, Cone, BC, and MG. Box plot showing the chromatin accessibility of the OCR harboring rs1328363 in 20 individuals with different genotype of rs1328363 in Rod. Scatter plot showing the WGS Ref/Alt read ratio (*X* axis) and snATAC-seq Ref/ Alt read ratio (*Y* axis) of rs1328363 in the individuals who are heterozygous for this variant. Violin plot showing the gene expression level of *GPC6* and *CRX* across retinal cell types (*n* = 50,000). CRX motif position weight matrix aligned to reference and alternative allele at rs1328363. A LocalZoom plot showing the refraction error and myopia GWAS signal, $-log_{10}(p-value)$, of rs1328363 and the surrounding SNPs. Genome track of *GPC6* locus showing cell type-specific chromatin accessibility and positioning of rs1328363 within an OCR, which is a predicted LCRE of *GPC6*

Moreover, our integrative analysis provided insights for cell type-specific regulatory mechanisms underlying GWAS loci (Fig. 8d). For example, rs1328363 is a fine-mapped variant associated with refraction error and myopia (PIP = 0.30). rs1328363 is also a sc-ASCA and nominally significant sc-caQTL of its residing OCR and a nominally significant sc-eQTL of *GPC6*, specifically in Rod. This variant may exert Rod-specific effect through strengthening the binding of a photoreceptor-specific TF, CRX, to a LCRE of *GPC6*, which is accessible in multiple cell types (Fig. 8d). Interestingly, this variant is

also a marginally significant sc-eQTL in Cone, concordant with CRX also being a TF for Cone. The lower expression of *GPC6* in Cone that aligns with lower accessibility of the corresponding LCRE and *GPC6* promoter in Cone, may contribute to the observed marginal sc-eQTL effect. Furthermore, *GPC6* encodes a putative cell surface glypican coreceptor with a potential role in controlling cell growth and division, implicating that rs1328363 may contribute to refraction error through inducing dysregulation in the growth and division of photoreceptor cells.

## Discussion

By integrating single-cell multiomics profiling, genomic sequencing, and ex vivo high-throughput reporter assay, we conducted a comprehensive study to investigate the impact of genetic variants on gene regulation in the human retina at cellular resolution. We identified regulatory elements, mapped effect of genetic variants, and elucidated modulation mechanisms underlying gene regulation in individual cell type contexts in vivo. The effects of genetic variants on gene expression measured by sc-eQTLs and sc-ASE are highly concordant, while the effects of genetic variants on chromatin accessibility assessed by sc-caQTLs and sc-ASCAs also show consistency. Additionally, sc-eQTLs are enriched of bulk eQTLs from retina and other tissue types, and higher overlapping rate was observed for sc-eQTLs identified in the most abundant cell type or those common in multiple cell types. Furthermore, by ex vivo MPRAs, a significant proportion of OCRs harboring sc-eQTLs exhibit regulatory activity in mouse retina, suggesting perturbation of these OCRs by genetic variants may result in gene expression change. Altogether, these results support the mapped genetic effects on gene expression and chromatin accessibility. Interestingly, a significant proportion (44.0%) of sc-eQTLs are not captured by bulk eQTLs, which might be due to most of the sc-eQTLs being cell type-specific, thus the cell type-specific signals, in particular those associated with cell types that represent a small proportion of the total retinal cell population, might be diluted and not detectable in the bulk level. It is also likely that some sc-eQTLs have opposite effect direction in different cell types, so the overall effect in the bulk level is canceled out, although we observed a very small fraction of sc-eQTLs in such cases.

   Intriguingly, the majority of sc-eQTLs and sc-caQTLs exhibit cell type-specific effects, whereas the eGene and caPeaks associated with these variants are expressed or accessible in multiple cell types, suggesting genetic variants modulate gene expression and chromatin state in a cell type-dependent manner. Surprisingly, the majority of cell type-specific sc-eQTLs and sc-caQTLs reside in the genomic regions accessible in multiple cell types. Furthermore, the TFs with motifs perturbed by genetic variants have higher expression level in the cell types where the variants exert cell type-specific effect, compared to the cell types where the variants do not have impact. Additionally, the enhancers validated by MPRAs are enriched of the binding of cell lineage/type-specific transcription factors and active enhancer epigenetic modifications, as shown by ChIP-seq analysis. Interestingly, the co-binding of an early lineage-determining factor and at least one late cell type/context-dependent factor was enriched in the validated enhancers, compared to the inactive sequences. Altogether, our study suggested that cell type specificity of genetic effect is driven by the precise control of TF expression and chromatin accessibility of cis-elements (Fig. 9). The specificity is achieved by perturbing the TF

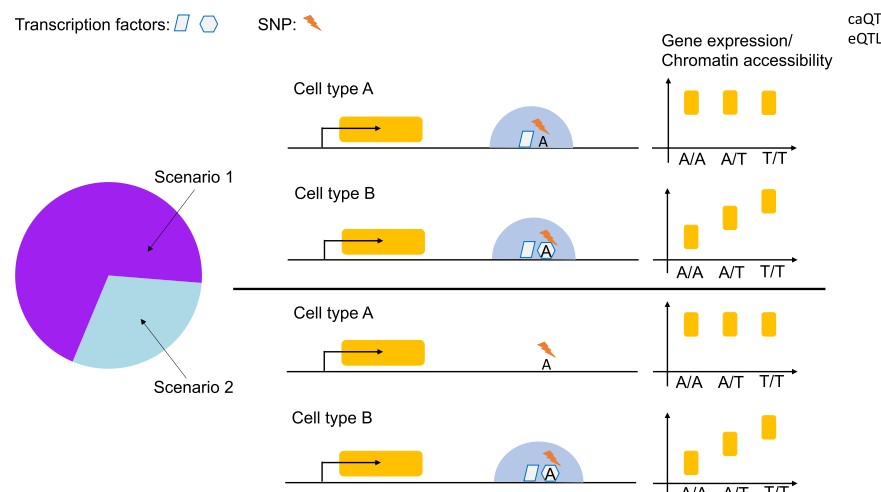

**Fig. 9** A model of the cell type-specific effect of genetic variants. The schematic plot shows that the cell type-specific effect of genetic variants could be driven by perturbing the binding of cell type-specific trans-regulators (Scenario 1), and/or attributed to cell type-specific chromatin accessibility of the associated cis-elements (Scenario 2), indicating hierarchical transcription factor collaboration may play important role in cell type-specific effects of genetic variants on gene regulation. For the cell types sharing a similar lineage, scenario 1 may account for a larger proportion of cases. However, for the cell types from different lineages, further investigation is needed to determine the contribution of the two scenarios

binding motifs that are actively involved. Specifically, we hypothesized hierarchical transcription factor collaboration may be crucial in gene regulation, whereby the pioneer factors first establish chromatin accessibility of the regulatory genomic regions in cells sharing a similar lineage, resulting in different cell types sharing common OCRs, then additional cell type/context-specific trans-factors may be recruited to these OCRs later in a collaborative manner to regulate gene expression. Therefore, genetic variants affecting the binding of cell type/context-specific trans-factors within the common OCRs could have cell type-specific effect on gene expression and chromatin accessibility. These results also suggested the accessibility of a genomic region does not necessarily indicate its activity, and an accessible regulatory element may be inactive and can be activated by the binding of additional trans-factors in a given cellular context. However, for the cell types from different lineages, the cell type-specific accessibility of affected cis-elements may play important role in determining the cell type specificity of genetic variant effects, which needs further investigation.

Moreover, we showed that integration of single-cell multiomics and GWAS studies can increase the resolution to nominate relevant cell context, causal variants and genes, and better dissect the underlying regulatory mechanisms. Specifically, the cell type enrichment of GWAS traits measured by gene expression and chromatin accessibility converged to the same patterns, supporting the reliability of our result, and suggesting some GWAS variants may indeed affect regulatory elements linked to gene expression in specific cell type context. Intriguingly, our analysis showed that astrocyte and MG may play important role in POAG, which was replicated in an independent study [49]. We also found MG may be involved in choroid/retina disorders, retinal detachments/breaks, and retinal problems, suggesting non-neuronal cell types, particularly glia cells, may be critical for neuronal diseases [50]. MG and astrocyte are macroglia cells in the retina and

play essential roles in maintaining the homeostasis and proper function of the retinal neurons [51]. In particular, astrocytes are located in the nerve fiber and ganglion cell layers, support the structure and physiology of the optic nerve head axon, and modulate the extracellular matrix under elevated IOP [52], supporting their important role in glaucoma. Furthermore, we observed myopia and refraction error might be involved most of retinal cell types, which was suggested by another independent study [53]. Additionally, we fine-mapped GWAS loci based on functional annotation of genetic variants, which prioritize the variants in regulatory regions as candidate causal variants. By overlapping the fine-mapped GWAS variants with sc-eQTL and LCREs, we identified the genes potentially contributing to myopia/refraction error and glaucoma. Moreover, combining gene expression, chromatin accessibility, and their variation driven by genetic variants in individual cell type contexts, we explained the cell type-specific regulation mechanism underlying GWAS loci, which could be related to cell type-specific trans-factor binding and chromatin state of cis-elements. These findings provide guidance for functional analysis of GWAS loci.

The specificity of genetic variant effects depends on the context in which the trans-factors with motifs perturbed by genetic variants operate. Specifically, if genetic variants disrupt the binding of pioneer factors, the associated cis-elements could become inaccessible, preventing the binding of additional cofactors. Consequently, diseases may onset in broader cell types or spatial/temporal contexts. In contrast, if genetic variants disrupt the binding of late or context-dependent trans-factors, the cis-elements could become accessible by pioneer factors but hinder the binding of these late or context-dependent trans-factors. Consequently, diseases may be triggered in specific spatial and temporal cellular contexts. Additionally, since the same TFs and cis-elements can be utilized in different cellular contexts, diseases may occur in multiple cellular contexts. In summary, disease onsets could be determined by the spatial and temporal cellular contexts where the disrupted TF motifs are actively involved. These insights could facilitate the understanding of pathogenic mechanisms and development of treatments for diseases.

One limitation of our study is the small cohort size, which limited our power to detect genetic effects such as sc-eQTLs and sc-caQTLs. Increasing the sample size and refining the mapping methods, such as applying a linear mixed effect model or modeling the correlation of effects in different cell type contexts, would significantly enhance the power to detect QTLs. Furthermore, differences in cell type proportions may also impact the detection power of genetic association studies. This is because accurate measurement of chromatin accessibility and allelic-specific effects relies on sufficient sequencing depth, which can be influenced by differences in cell type proportions. Additionally, the low capture rate of RGCs and astrocytes per individual due to cell type proportion differences limited their inclusion in the genetic association studies. This compromised our ability to effectively fine-map primary open-angle glaucoma (POAG) by overlapping GWAS variants with sc-QTLs, as RGCs are the affected cell type in glaucoma. We anticipate that enriching for cell types representing a small proportion of the total retinal cell population will increase the power to map QTLs, identify allele-specific variants, and nominate causal variants and genes for GWAS loci. Another potential improvement is to simultaneously profile chromatin accessibility and gene expression within the same cell

using single-cell multiome sequencing. This approach would allow for matching chromatin accessibility and allelic expression within individual cells, thereby increasing the power to validate our findings and gain further insights into gene regulation.

## Conclusions

We conducted the first systematic study to investigate how common genetic variants modulate gene expression and chromatin accessibility in major cell types of the human retina through integrative single-cell multiomics analyses. Our findings highlight the context-dependent nature of genetic variant effects on gene regulation. Many genetic variants reside in genomic regions accessible across multiple cell types but exert effects only in specific cell types. These observations suggest hierarchical collaboration among transcription factors plays a crucial role in cell type-specific effects of genetic variants on gene regulation. Our comprehensive study provides novel insights into the nucleotide-level mechanisms of gene regulation at cellular resolution, shedding light on understanding and treating human diseases.

## Methods

### Human retina sample collection

Samples included in this study were retinal tissues of 20 donors from the Utah Lions Eye Bank (Additional file 2: Table S1). All donors were screened for medical history, and only the ones with no records of retinal diseases were used in this study. Postmortem phenotyping with OCT were performed to confirm that there were no drusen, atrophy, or any other disease phenotypes on retina by our previous approach [54]. One eye was collected from each donor. All eye tissues were collected and dissected within 6 h postmortem, according to a previous protocol [55]. With 4 and 6 mm disposable biopsy punches, macula and peripheral retina were collected and flash-frozen in liquid nitrogen, and stored at $-80\ ^\circ$C before nuclei isolation.

### Nuclei isolation and sorting

Nuclei for snRNA-seq were isolated by fresh-made pre-chilled RNase-free lysis buffer (10 mM Tris–HCl, 10 mM NaCl, 3 mM MgCl2, 0.02% NP40). The frozen tissue was resuspended and triturated in lysis buffer and homogenized with a Wheaton™ Dounce Tissue Grinder. Isolated nuclei were filtered with a 40-μm Flowmi Cell Strainer. DAPI (4′,6-diamidino-2-phenylindole, 10 μg/ml) was added before loading the nuclei for fluorescent cytometry sorting with a BD (Becton Dickinson, San Jose, CA) Aria II flow sorter (70 μm nozzle). The sorted nuclei are ready for snRNA-seq.

Nuclei for snATAC-seq were isolated in fresh-made pre-chilled lysis buffer (10 mM Tris–HCl, 10 mM NaCl, 3 mM MgCl2, 0.02% NP40, 1% BSA). Similar to the nuclei isolation process for snRNA-seq, frozen tissue was homogenized with a Dounce Tissue Grinder until no tissue pieces were visible. Nuclei were then washed (wash buffer: 10 mM Tris–HCl, 10 mM NaCl, 3 mM MgCl2, 1% BSA) twice in a pre-coated (coating buffer: 10 mM Tris–HCl, 10 mM NaCl, 3 mM MgCl2, 4% BSA) 5 ml round-bottom Falcon tube (Cat. NO. 352,054) at 500 g, 4℃ for 5 min. Nuclei were resuspended in $1 \times$ diluted nuclei buffer (10X PN-2000153, PN-2000207) for a final concentration of 3000–5000 nuclei/μl.

### Single-nuclei sequencing

Single-cell Gene Expression Library was prepared according to Chromium Next GEM Single Cell 3' Reagent Kits v3.1 (10 × Genomics). In Brief, single-nuclei suspension, reverse transcription (RT) reagents, Gel Beads containing barcoded oligonucleotides, and oil were loaded on a Chromium controller (10 × Genomics) to generate single-cell GEMS (Gel Beads-In-Emulsions) where full-length cDNA was synthesized and barcoded for each single cell. Subsequently, the GEMS are broken and cDNA from each single cell are pooled. Following cleanup using Dynabeads MyOne Silane Beads, cDNA is amplified by PCR. The amplified product is fragmented to optimal size before end-repair, A-tailing, and adaptor ligation. Final library was generated by amplification. After quantification with KAPA Library Quantification kit (Roche), libraries were sequenced on a Novaseq 6000 Sequencer (Illumina).

Single-cell ATAC Library was prepared according to Chromium Next GEM Single cell ATAC Reagent kit v1.1 (10 × Genomics). In Brief, prepared nuclei were incubated with transposome and transposase entered and preferentially fragmented DNA in open region of chromatin. Transposed single nuclei, a master mix, Gel Beads containing barcoded oligonucleotides, and oil were loaded on a Chromium controller (10 × Genomics) to generate GEMS (Gel Beads-In-Emulsions) where barcoded single strand DNA was synthesized. Subsequently, the GEMS are broken and pooled. Following sequential cleanup using Dynabeads MyOne Silane Beads and SPRI beads, barcoded DNA fragments are amplified by PCR to generate indexed library. After quantification with KAPA Library Quantification kit (Roche), libraries were sequenced on a Novaseq 6000 Sequencer (Illumina).

### Whole genome sequencing

One microgram genomic DNA was sheared with Covaris for 70 s and the purification was performed with Ampure XP beads. After end repair and A-tailing, the indexed adaptors were added to the product, and subsequently purified with Ampure XP beads. The diluted library was then sequenced in an Illumina Novaseq6000 Sequencer.

### WGS data processing

The WGS variant calling followed the GATK pipeline for analyzing small sample cohorts (https://gatk.broadinstitute.org/hc/en-us/articles/360035890411-Calling-variants-on-cohorts-of-samples-using-the-HaplotypeCaller-in-GVCF-mode). Briefly, WGS data was aligned to the human reference genome (build hg19) with BWA-MEM [56]. After removing duplicate reads with MarkDuplicates (Picard) from GATK, the bam files were realigned with base quality score recalibration and local realignment with GATK4 [57]. With the realigned bam files, the variants were called to generate genome-wide genotype-per-site data for each sample (gVCF). The joint genotyping was performed on variants of all samples using GATK GenotypeGVCFs. Variants from joint genotyping underwent variant recalibration with GATK.

### The QC and filtering of the WGS variants

QC and filtering of the WGS variants were performed with the following criteria: (1) Indicated as "PASS" by GATK based on the filtering criteria: For SNP: $QD < 2.0$,

QUAL < 30.0, SQR > 3.0, FS > 60, MQ < 40, MQRankSum < -12.5, ReadPosRankSum < -8.0. For INDEL: QD < 2.0, QUAL < 30.0, FS > 200, ReadPosRankSum < -20.0. (2) Remained dimorphic after assigning the variants as missing with the following criteria: (a) the variants not called; (b) variant calls with allelic imbalance (AB > 0.9 or AB < 0.1); (c) variant GQ per sample (from GATK) < 20; (d) heterozygous genotype calls on chrX nonPAR of the male donors; (e) heterozygous genotype calls on chrY nonPAR of the male donors or, heterozygous/homozygous calls on chrY nonPAR of female donors. (3) A missingness rate ≤ 20%. (4) Genomic sites not involving MNP. (5) Not in low complex regions. (6) Passing Hardy–Weinberg Equilibrium testing. The QC and filtering were performed using GATK, plink, bcftools, and custom scripts in Perl [57, 58]. After QC and filtering, a total of 9,792,238 variants were obtained for downstream analysis.

### Quality control of sample genotypes

The sample genotypes were QC using multidimensional scaling (MDS) analysis of plink with the genotype data from the Hapmap project [58, 59] (including 84 CHB individuals, 117 CEU individuals, 115 YRI individuals). Briefly, the MDS analysis was performed with the filtered autosomal SNPs that were presented in both donors and Hapmap populations. The 20 samples were clustered with the Hapmap populations based on the MDS analysis, which is consistent with the reported ethnicity of these samples, including 16 European, 3 Latino, and 1 Asian (Additional file 2: Table S1, Additional file 1: Fig. S2a).

### Phasing with reference panel

The SNPs aligned between the 1000 genome phase 3 reference panel and the genomes of the 20 samples were extracted with shapeit2 [60, 61]. For each autosome, the overlapped SNPs of the sample genomes were phased with shapeit2 using the reference panel haplotypes with the same ethnicity as the sample group.

### snRNA-seq processing

The snRNA-seq raw data were processed with cell ranger. To remove the ambient RNA contamination, the read count matrix (gene × cell) was corrected with SoupX for each sample [62]. For each sample, to remove low-quality cells, the corrected count matrix was filtered using the following parameters: min.cells = 5, nFeature_RNA ≥ 500, percent. mt ≤ 15 by Seurat [63]. To remove doublets, DoubletFinder was applied to each sample with doublet rate set at *thecellnumber*/1000 × 0.01 [64]. After removing doublets for each sample, cell types were predicted using scPred based on the reference retinal cell atlas [27, 65]. The expression of marker genes per cell type per sample were examined to confirm cell type assignment. For all the violin plots presented in the "Results" section, we utilized 50,000 cells down-sampled from the original snRNA-seq dataset ($n = 192,792$).

### snRNA-seq gene expression quantification

For each cell type, the average CPM of each gene across the cells from the same cell type of a sample was computed as the gene expression measurement per sample. For each cell type, the gene expression of all genes in the 20 samples were collected (gene × sample matrix) to perform quantile normalization. For each gene per cell type, the normalized

gene expression levels were transformed using rank based inverse normal transformation [66]. For each cell type, only the genes with mean CPM (in the 20 samples) ≥ 5 were kept for downstream sc-eQTL analysis. To remove the effects of confounding variables (e.g., batch effect) from gene expression, the PEER factors were calculated from the transformed gene expression with the "PEER" R package [67, 68].

### snATAC-seq processing

The snATAC-seq raw data were processed with cell ranger and then analyzed with ArchR [69]. The QC and filtering of low-quality cells and doublets were performed with ArchR using the default setting (minTSS = 4 and minFrags = 1000, doublet filter-Ratio = 1). The cell types of snATAC-seq were assigned by integrating the snRNA-seq data of the 20 samples using ArchR. For each sample, the snATAC-seq bam file per cell type per donor was generated according to the cell type label. For each cell type, the bam files from the same cell type of the 20 donors were merged to call snATAC peaks with macs3 in the default setting [70]. To reduce false positive peaks, only the peaks with mean FPKM ≥ 2 across samples per cell type were kept for each cell type. The filtered peaks from all cell types were combined to generate a set of standardized peak coordinates that can be compared among different cell types using the "Reduce" function in R. The peaks in the hg19 blacklist regions (wgEncodeHg19ConsensusSignalArtifactRegions) and chrY were filtered out. The standardized peak set was input into ArchR to generate peak to gene connection list, peak co-accessibility list, and the differential accessibility regions (DARs). The TFs were identified from the OCRs per cell type by chromVAR and correlating the TF expression with their motif enrichment across cell types (p.adj < 0.01, correlation coefficient > 0.5, and a maximum inter-cluster difference in deviation $z$-score > 75th percentile) with ArchR.

### sc-eQTL mapping

For each cell type, cis-eQTLs were mapped for the genes with mean CPM ≥ 5 using FastQTL [71]. Only the variants passing the following criteria were considered: (1) within ± 250 kb of gene TSS, (2) in OCRs of the given cell type, (3) with minor allele frequency (MAF) ≥ 0.1 across the 20 samples, and (4) minimum number of samples carrying the minor allele ≥ 4. Given the small sample size ($N = 20$), three PEER factors and the first component of MDS analysis of the genotypes were used as covariates. Given our relatively small cohort of $N = 20$, we include the top three PEER factors and the first component of MDS analysis of the genotypes as covariates, to balance the need for confounding correction while preserving the detection power for eQTL analysis (see "The selection of covariates for eQTL mapping" section in Additional file 1: Supplementary Note). The FastQTL were run in a nominal pass mode. To identify gene-level significant sc-eQTLs, the $p$-value of each sc-eQTL per gene was corrected for multiple testing with Bonferroni method, based on the number of independent variants per gene estimated by eigenMT [72], for each cell type respectively.

### sc-ASE mapping

The snRNA-seq bam file per cell type per sample was generated according to the cell type label. To correct read mapping bias, the snRNA-seq bam file per cell type per sample

were processed with WASP [73]. Duplicate reads were removed with UMI-tools [74]. For each sample, the reference panel phased SNP VCF and corrected snRNA-seq bam files were used to generate haplotype count and genome-wide phased VCF with phASER [75]. The gene-level haplotype counts for allelic expression were obtained using phASER Gene AE. For each cell type, the gene-level haplotypic counts per sample were combined to produce a haplotypic expression matrix (gene × sample) using phaser_expr_matrix. py of phASER-POP [76]. For each cell type, the effect sizes of all tested variant-gene pairs in the aforementioned sc-eQTL analysis were calculated using the aggregated haplotypic expression matrix and the genome-wide phased VCF with phaser_cis_var. py of phASER-POP. Only the variants with ≥ 4 heterozygotes are considered. For each cell type, genome-wide multiple testing correction was performed for each variant with Benjamini–Hochberg method. The variants with FDR < 10% were identified as sc-ASEs.

### sc-ASCA mapping

For each sample, to correct read mapping bias, the snATAC-seq bam file per cell type per sample were processed with WASP [73]. Duplicate reads were removed with Mark-Duplicates (Picard) from GATK [57]. The allelic count of SNPs was obtained using ASEReadCounter from GATK. For each SNP per cell type per sample with at least 10 reads from WGS and 10 reads from snATAC-seq are considered, and a one-sided Fisher test was used to compare whether the allelic count ratio from snATAC-seq was significantly greater or less than the allelic count ratio of from WGS. For each cell type, the Fisher test *p*-values of the same SNP in all heterozygous samples were combined to calculate a meta *p*-value using the Stouffer's method with the "metap" R package [77] (with the total read count in WGS-seq and snATAC-seq as the weight for each sample). For each SNP per cell type, only the meta *p*-value in the effect direction with more significance was kept. For each cell type, the SNPs passing the follow criteria were considered: (1) with at least of three heterozygous samples and (2) considered in the aforementioned sc-eQTL analysis. To correct for genome-wide multiple testing, for each cell type, Benjamini–Hochberg correction was applied to meta *p*-value of each SNPs to identify sc-ASCAs with FDR < 10%.

### sc-caQTL mapping

For each cell type, the fragment count matrices (peak × sample) were generated based on the standardized peak coordinates in the given cell type and the snATAC-seq bam file (after WASP correction and removal of duplicate reads) per sample per cell type using featuerCounts [78]. For each cell type, the reference panel phased SNPs were annotated with allelic read counts using RASQUAL tools [79]. To correct for library size and GC content bias in feature-level fragment counts per sample, the sample-specific offset was computed using the rasqualCalculateSampleOffsets() function with the "GC-correction" option. The fragment count covariates were calculated with make_covariates() function of rasqual package (with variable number of covariates in different cell types) and were included into the model. For each cell type, sc-caQTL analysis was performed for the variants that were considered in sc-eQTL analysis. RASQUAL was run in two modes: (1) in the default setting and (2) with permuted sample labels using the "—random-permutation" option. To correct for multiple testing in feature level, the number of independent

variants/tests per peak was determined with eigenMT [72]. Based on the number of independent tests, the true association *p*-values and empirical permuted *p*-values were corrected with Bonferroni method respectively. To correct for multiple testing genome-wide, the corrected true association *p*-values were compared to the corrected empirical null distribution to determine the true *p*-value threshold with FDR < 10%.

### LCRE identification

The gene-peak links were identified based on the correlation of gene expression and chromatin accessibility of snATAC-seq OCRs ($\pm$ 250 kb) using the addPeak2GeneLinks() function in ArchR [69] with binarized peak read counts. The peak co-accessibility was estimated with the addCoAccessibility() function in ArchR with binarized peak read counts (for OCRs in $\pm$ 250 kb). The snATAC-seq OCRs were annotated with ChIPseeker [80] and the OCRs within $\pm$ 1 kb surrounding the promoter regions were defined as promoters. From the gene-peak links, we selected the OCRs that are not promoters as the CREs of the linked genes, while from the peak co-accessibility links, we selected the OCRs linked to promoters as the CREs of the target genes. The union set of gene-peak links and peak co-accessibility links were defined as the linked cis-regulatory elements (LCRE) of the associated genes.

### Predicting the motif disrupting effects of SNPs

To determine if genetic variants within OCRs affect TF binding sites (TFBSs), we identified known TF motifs to the sequence surrounding genetic variants with motifBreakR [81], based on 2817 TF motifs (Hsapiens) from MotifBreakR database. The relative entropy of the motifs with reference allele and alternative allele was calculated, and only the TFBSs that were strongly affected (effect = "strong") by SNPs were considered (with the parameters: filterp = TRUE, threshold = 1e − 4, method = "ic"). We further required a TF with CPM $\geq$ 50 in a cell type to determine if its motif is perturbed by genetic variants in the corresponding cell type, as TF motifs were prevalent in the genome and including TFs with low expression in the corresponding cell type may bring false negatives and increase the background noise.

In our analysis, we examined the expression of transcription factors (TFs) whose binding sites are perturbed by genetic variants. Specifically, we compared their expression in cell types where the variants have an impact versus cell types where the variants do not exert any effect. To determine if a TF's motif is perturbed by genetic variants in a particular cell type, we first applied a criterion of TFs with CPM $\geq$ 50 in that cell type, then we also tested a criterion of TFs with CPM $\geq$ 20 in that cell type. As a result, our findings with both criteria consistently aligned with each other. The results with the criterion of CPM $\geq$ 50 were present in the "Results" section.

### Identification of LD-independent sc-caQTL and LD-independent sc-eQTL

PLINK v1.90b5.2 [58] (with the parameters: −clump-p1 0.05 −clump-p2 0.05 −clump-r2 0.50 −clump-kb 250) was used to clump sc-eQTLs per eGene per cell type and to clump sc-caQTLs per caPeak per cell type. The SNPs with the smallest *p*-value were assigned as the index SNPs. For multiple index SNPs with the same *p*-value, the SNP that is closest

to gene TSS was assigned as the index sc-eQTL SNP, while the SNP that is closest to peak summit was assigned as the index sc-caQTL SNP.

### Identification of caQTLs associated with multiple genomic regions

For each common variant within snATAC-seq OCRs, we tested the association between the variant and the accessibility of snATAC-seq OCRs in $\pm 250$ kb surrounding the variant and took $p < 0.005$ as significant association. If the variant itself is a sc-caQTL of its residing OCR and also associated with other surrounding OCRs, we defined it as a sc-caQTL associated with multiple genomic regions and the residing OCR as the master caPeak while the other surrounding peaks as the dependent caPeaks. To avoid the confounding effect that two sc-caQTLs affecting two master caPeaks are in LD, the OCR that is a master caPeak and has its own resident caQTL that is in LD with the tested variant ($r^2 > 0.5$) was filtered out.

### Massively parallel reporter assays (MPRAs)

We developed a MPRA library that contains the sequences of the 931 open chromatin regions harboring the index sc-eQTLs identified in the rod cells, along with 20 control cis-regulatory elements (CREs) with a variety of activity that have been previously validated [35] and negative controls (i.e., 300 scrambled sequences, and a basal promoter without CRE). Each CRE or control sequence was labeled with three unique barcodes. The basal promoter was assigned 25 barcodes. Oligonucleotides (oligos) were synthesized as follows: 5′ priming sequence /EcoRI site/Library sequence (224-bp)/SpeI site/C/SphI site/Barcode sequence (9-bp)/NotI site/3′ priming sequence, and ordered from TWIST BIOSCIENCE (South San Francisco, CA). These oligomers were cloned upstream of the photoreceptor-specific Crx promoter that drives the expression of a DsRed reporter gene. The resulting plasmid library was electroporated into 3 retinal explants of C57BL/6 J mice at postnatal day 0 (P0) in four replicates. At day 8, DNA and RNA were extracted from the cultured explants and underwent next-generation sequencing. The activity of each CRE was calculated based on the ratio of RNA/DNA read counts and was normalized to the activity of the basal Crx promoter. The bioinformatics analysis of the MPRA followed the pipeline previously published, with R and Python scripts [35].

### Cell type enrichment of GWAS loci

To determine the cell type enrichment of GWAS loci, we analyzed chromatin accessibility and gene expression derived from single-cell multiomics data respectively. For chromatin accessibility, we partitioned the heritability of GWAS traits into the cell type OCRs and DARs through stratified LD score regression based on the summary statistics of GWAS traits with LDSC [43]. First, for each GWAS study, the SNPs that are overlapped with HapMap3 SNPs were annotated based on whether they are in the OCRs or DARs in each cell type. Then, LD-scores of these SNPs were calculated within 1 cM windows based on the 1000 Genome data. For each GWAS study, the LD-scores of these SNPs were integrated with the ones from baseline model, which include non-cell type-specific annotation, downloaded from https://alkesgroup.broadinstitute.org/LDSCORE/. Finally, for each GWAS study, the heritability in the annotated genomic regions were estimated

and compared with the baseline model to determine if the genomic regions in each cell type were enriched with the heritability of the corresponding GWAS trait. The enrichment *p*-values were further corrected to compute q-values using "qvalue" R package given the number of cell types and GWAS studies considered.

For gene expression, we assessed whether there is linear positive correlation between cell type specificity of gene expression and gene-level genetic association with GWAS studies by MAGMA.Celltype [44–47]. We formatted GWAS summary statistics with the "MungeSumstats" R package, included SNPs $-35$ kb/$+10$ kb of each gene based on 1000 genome "eur" population, formatted snRNA-seq expression data with the "EWCE" R packages, and detected the linear enrichment with MAGMA.Celltype [44–47]. The enrichment *p*-values were further corrected to compute q-values using "qvalue" R package given the number of cell types and GWAS studies considered [43–47].

### Fine-mapping GWAS loci

We fine-mapped GWAS loci based on the summary statistics of GWAS studies [36, 40, 42, 48]. For each GWAS study, the SNPs with $p < 5 \times 10^{-8}$ and present in 1000 genome (phase 3) European population were considered and were divided in the LD blocks identified by previous study [82]. The prior of each SNP was computed based on GWAS *Z*-score and the functional annotation of the SNP with "TORUS" package [83]. The annotation of a SNP was assigned to one of the categories: "4" if the SNP in the exonic/UTR regions, "3" if the SNP in the promoter region, "2" if the SNP in LCRE, "1" if the SNP in snATAC-seq OCR, "0" if the SNP not in snATAC-seq OCR. For each LD block, we calculated the PIP of each SNP and credible set of SNPs with the aforementioned prior weight generated by TORUS (i.e., functional PIP) and without the weighted prior (uniform PIP), respectively with "susieR" package [84]. Then we overlapped the fine-mapped variants with functional PIP > 0.1 with sc-eQTL, sc-caQTL and sc-ASCA.

### The colocalization of GWAS loci with retina bulk eQTL

We identified SNPs with GWAS analysis $p < 10^{-6}$, focused on $\pm 500$ kb surrounding the SNP, and considered the genes whose TSS are in the region. Then we extracted the GWAS variants and retinal bulk eQTL variants [85] of the genes in the region and performed colocalization using the "coloc" R package (v5.1.0) with default parameter setting based on the summary statistics of the GWAS study and retinal bulk eQTLs for each of the corresponding genes respectively [86, 87]. We identified the genes with the posterior probabilities of colocalization (PP4) > 0.5 as the genes colocalized with GWAS signals [22].

### Processing of the bulk ATAC-seq and ChIP-seq data

The human retina and macula bulk ATAC-seq and bulk ChIP-seq of OTX2, CRX, NRL, MEF2D, CTCF, H3K27ac, and H3K4me2 data were downloaded from https://www.ncbi.nlm.nih.gov/geo/query/acc.cgi?acc=GSE137311 [28]. The sequencing reads were trimmed with Trimmomatic-0.39 and mapped to the hg19 genome assembly with bowtie2-2.3.5.1 (The alignment bam files were sorted by samtools-1.2) [88–90]. The narrowPeaks were called from the bam files with Genrich (https://github.

com/jsh58/Genrich). The narrowPeaks from the bulk ATAC-seq and ChIP-seq were then intersected with snATAC peaks with bedtools [91].

## Supplementary Information

---

**Additional file 1: Fig. S1.** The OCRs identified from snATAC-seq. **Fig. S2.** Identification of sc-eQTL. **Fig. S3.** Identification of sc-caQTL. **Fig. S4.** Identification of sc-caQTL in down-sampled cells. **Fig. S5.** Cell type enrichment and fine-mapping of GWAS loci. **Fig. S6.** An example of the fine-mapped GWAS candidate variants with the retinal bulk eQTL signal of its target gene colocalized with GWAS signal. Supplementary Note. Acronyms list.

**Additional file 2: Table S1.** The donor retina sample information. **Table S2.** The cell number per cell type per donor from snRNA-seq and snATAC-seq. **Table S3.** snATAC-seq OCR list. **Table S4.** The summary statistics of sc-eQTL with gene level FDR < 0.1 per cell type. **Table S5.** The effect size of sc-eQTL in significant retinal cell types. **Table S6.** The distance between eGene TSS and sc-eQTL in significant retinal cell types. **Table S7.** The gene expression level of transcription factors with motifs perturbed by cell type-specific sc-eQTLs. **Table S8.** The summary statistics of sc-caQTL with genome level FDR < 0.1 per cell type. **Table S9.** The effect size of sc-caQTL in significant retinal cell types. **Table S10.** The gene expression level of transcription factors with motifs perturbed by cell type-specific sc-caQTLs. **Table S11.** The master and dependent regions of sc-caQTLs across retinal cell types. **Table S12.** The regulatory activities of sequences tested with MPRAs. **Table S13.** The fine-mapped GWAS variants with PIP >0.1 and overlapped with sc-QTL and/or sc-ASCA. **Table S14.** The target genes of the prioritized GWAS variants with PIP >0.1 and overlapped with sc-QTL and/or sc-ASCA.

**Additional file 3.** Review History.

---

### Acknowledgements

We acknowledge the computing cluster server in the Department of Molecular and Human Genetics and Human Genome Sequencing Center at Baylor College of Medicine for providing the computing resource. We thank Michael Casey in the DOVS Molecular Genetics Service Core, Washington University for his technical support in constructing plasmid MPRA libraries.

### Review history

The review history is available as Additional file 3.

### Peer review information

### Authors' contributions

RC, JW, XC, YL, and SC conceived the study and designed experiments. MD provided human samples. XC, YL, LO, JL, and YZ performed the experiments. JW, QL, and MW analyzed the data. JW, RC, XC, YL, and SC wrote the manuscript. All the authors edited and approved the manuscript.

### Funding

This study was supported by the Foundation Fighting Blindness (BR-GE-0613–0618-BCM), the National Eye Institute (R01EY022356, R01EY020540, R01EY018571) and Human Cell Atlas Seed Network Grant CZF2019-02425 to RC. This work was performed at the Single Cell Genomics Core at BCM partially supported by NIH shared instrument grants (S10OD023469, S10OD025240), P30EY002520 and CPRIT grant RP200504. This work was also supported by Vision Core Grant P30 EY002687 to Washington University.

### Availability of data and materials

The codes for reproducing the main analyses of the paper are available at GitHub (https://github.com/fe4960/single_cell_QTL) [92]. The data of 20 macular snRNA-seq samples, 20 lobe snATAC-seq samples and 1 macular snATAC-seq sample generated in the study are available with the accession number GSE247157 [93] and the data of 19 macular snATAC-seq samples analyzed in this study are available with the accession number GSE226108 [94] in Gene Expression Omnibus (GEO) under NCBI.

## Declarations

### Ethics approval and consent to participate

All tissues were de-identified under HIPAA Privacy Rules. Institutional approval for the consent of patients for their tissue donation was obtained from the University of Utah and conformed to the tenets of the Declaration of Helsinki. The IRB approval number is H-42822.

### Consent for publication

Not applicable.

### Competing interests

The authors declare that they have no competing interests.

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

## 