## [**Additional file 3.** Review History. · Genome Biology]

Review History

First round of review

Reviewer 1

Are you able to assess all statistics in the manuscript, including the appropriateness of statistical tests used? Yes, and I have assessed the statistics in my report.

Comments to author:

Wang et al provided a comprehensive single-cell multiomic analysis of major cell types of the human retina. They mapped eQTL and caQTLs which were well supported by allele-specific expression and chromatin accessibility, and discovered new QTLs missed by earlier studies using bulk sequencing. Importantly, they observed many cell type-specific genetic regulatory effects and presented thought-provoking analyses implicating a sophisticated transcription factor network underlying cell type specificity. Lastly, they performed a well-designed fine-mapping analysis to identify potential causal variants and relevant cell types. Even though some details were currently missing, the statistical analyses were appropriate and supported the authors conclusions. This work is highly relevant and very appropriate for Genome Biology. I have the following comments and questions for the authors.

- The authors omitted some important details on the identification of QTL in their data. In particular, why did they choose to include $N = 3$ PEER factors and 1 genotype PC? Did top PEER factors correspond to known technical covariates? What would happen to the power to detect eQTL if more PEER factors were included? How about other demographic variables such as sex and age?
- In addition, for caQTL mapping, what is the window for SNPs to be considered for each peak? Did the authors distinguish caQTLs where the SNP overlap with the peak, and those that are outside? (Maybe that's the "distal caQTLs" they mentioned in line 248, page 10, but the description of this distinction was not easily found).
- The authors observed that variants in wider OCRs have local effects, while the variants in narrow OCRs are more likely to affect accessibility of the entire OCR. This is interesting, although a confounding observation can be that the wider peaks called are less accurate (may be a union of multiple smaller peaks). What is the distribution of peak lengths in this dataset? An alternative approach for caQTL calling is to use the "tiled" peaks (defined as 500bp windows genome-wide in ArchR) and then map significant caQTLs to OCRs. Have the authors tried this approach of mapping caQTLs?
- The authors made the interesting observation that some cell-type specific caQTLs are associated with caPeaks which are accessible across multiple cell types. How did the authors address the possibility that there may be cases where the SNP is actually a caQTL in multiple cell types but did not pass statistical threshold because of differences in power, as a result of cell type abundance differences?
- The calculation of LD was not clearly presented. Given that this cohort has individuals from diverse ancestries, which reference panel was used to assess LD?

- In Figure 3F, the authors noted that the expression levels of TFs disrupted by eQTLs in the relevant cell types have higher expression than non-eQTL cell types. This is a very interesting observation and a convincing analysis to justify their arguments that trans factors may drive cell type-specific caQTLs. How many TFs were included in this analysis (predicted to have motifs perturbed?) Is there evidence from published studies supporting some of the TFs in establishing cell type specificity?

- The enrichment test of caQTLs associated with dependent OCRs in Figure 6A is interesting. How were the background "all tested variants" defined? It seems that the authors identified variants which are not caQTLs in the master peak but still are associated with distal OCRs (e.g. ~10% in Rod). Could the authors provide insights on this class of QTLs?

- In Figure 6C, please indicate what statistical tests were performed (compared to all tested OCRs or comparing between master and dependent OCRs). Similarly, the p-values should be reported in the text when comparing master and dependent OCRs in active chromatin (H3K27ac and H3K4me2).

- As a related question, could the authors provide more information on the relationship between master and dependent OCRs? Are they likely co-accessible peaks from scATAC, or is there relevant Hi-C data supporting their interactions?

- It is very interesting to compare the caQTLs affecting multiple peaks in concordant and opposite directions, and the authors presented convincing data supporting compensatory roles of OCRs in leading to cis-eQTLs. Again, is there evidence (from co-accessibility or Hi-C, for example) supporting their interactions in case they jointly determine a cis-eQTL?

- Among a total of 818 fine-mapped variants from three GWAS, the authors identified 27 QTLs in their dataset. Is this fraction higher than expected? (What fraction of GWAS SNPs were tested for QTLs in this study?) Further, could the authors provide a comparison of fine-mapping using existing data from bulk eQTL and annotated cis-regulatory regions, and demonstrate the gain in power to nominate potential causal variants in these traits, using single-cell multiomics and single-cell QTL mapping?

- The authors may consider clarifying the concept of "hierarchical transcription factor" network they proposed in this work, as it is not obvious what they refer to. Did they want to highlight the potential role of pioneer transcription factors, which establish chromatin accessibility and recruit other cell-type specific factors?

- I would suggest that the authors make changes to some figure legends to describe the data rather than interpret them. For example, in Figure 2g, what are the columns and rows of the heatmap? As another example, in Figure 4f-g, please describe the axes in figure legends and move the interpretation to the main text.

- The authors should consider using language more consistent with the current framework for describing population characteristics; for example, the use of the term "Caucasian" should be carefully avoided based on recommendations by the National Academies (<https://nap.nationalacademies.org/catalog/26902/using-population-descriptors-in-genetics-and-genomics-research-a-new>)

- Some references were not not correctly labeled. For example, in page 14, the authors wrote that "We incorporated functional annotation (including OCR and LCRE derived from single cell multiomics data) of variants to prioritize GWAS loci[48]56" but neither of the references matched their described methods.

Reviewer 2

Are you able to assess all statistics in the manuscript, including the appropriateness of statistical tests used? No, I do not feel adequately qualified to assess the statistics.

Comments to author:

In the manuscript by Wang et al., the Authors utilize single-cell RNA-sequencing, separate single-cell ATAC-sequencing, and whole genome sequencing of 20 healthy patient retinas to examine how genetic sequence variability affects chromatin structure and gene transcription within individual retinal cells. This large body of work provides a significant dataset where precise genetic information can be correlated with epigenetic and transcriptomic features within the mature retina, providing hypothesis-driving insights for mechanisms by which genetic variability results in phenotypic consequences.

The manuscript presents a heroic integration of a mountain of data, but suffers from major deficiencies.

1) The manuscript in its current form is an accumulation of correlated results between genetics, chromatin accessibility, and transcription, resulting in a broad survey of results listed as '% X of Y shows enrichment of Z'. The Authors do show specific examples of results (Figure 3d-e; 5a-b; 6d-7d) but the biological significance of any of these findings is not examined. The model (Figure 8) is built on the assumptions that transcription factor affinity is changed. While this may in fact be the correct conclusion, any reference to binding affinities of transcription factors is inferred based on canonical motifs, but not actual data.

2) The use of acronyms throughout the manuscript makes the dialog almost impossible to follow. This reviewer struggled to read most of the document, needing a reference table of acronyms to try and decipher what the Authors were trying to analyze and resulting conclusions from the analyses.

3) Concerns about the significance of analyses utilizing snRNA-seq with snATAC-seq
a. Ideally this would have been done in the SAME cells using the multiome kit so that chromatin accessibility, and allelic expression could be matched within individual cells. It is recognized that these experiments, however, were started prior to the availability of the multiome kit. Authors should state the limitations of the analysis because of this, especially when differences in cell type proportions captured from individual patients (see point b) are significant.

b. Throughout the manuscript, the authors examine cell-type specific patterns of expression/accessibility, but fail to include the retinal ganglion cells (RGCs). This is likely because capture rate of the RGCs is extremely low in the snATAC (Figure 1). All conclusions of cell-type specificity are therefore limited because a major and disease-relevant cell type is not included. There is no mention of the limitations and/or biases of the analysis when missing this cell type. Conversely, when the Authors examine relevant diseases (Figure 7), the RGCs are now included because primary open angle glaucoma is a disease where RGCs are affected. As the dataset is likely under-powered for RGC analyses, it seems disingenuous to state that RGCs don't have genetic features (accessibility/expression) that are likely to contribute to POAG.

c. snRNA-seq seems to show multiple rod populations, including a mixing of rods with many other cell types (bipolars, Mullers, others?). Potential causes of this variability is high amounts of ambient RNA (concern of snRNA-seq) and/or doublets. The methods state that 'SoupX' and 'DoubletFinder' were applied to remove doublets, but a low doublet threshold was utilized (0.01). The standard doublet rate for capturing 10,000 cells/run on a 10x Genomics v3.1 3' is 8% (0.08 doublet threshold).

d. Would be helpful to have UMAPs of scRNA and scATAC colored by sample to observe potential batch effects

4) Figure Legends - Most figure legends are interpretations of the data, not an actual description of the graphs, presented data. Here are two examples, but this is how the legends of a MAJORITY of the figures read.

Example 1:

Figure 5b legend (Figure 5c is also labeled as 5b) - "The variant rs6859300 is a Rod-specific sc-caWTL of its residing OCR, and resides in an OCR accessible in ROD, Cone, and BC. This variant is predicted to enhance binding of EPAS1, increasing chromatin accessibility of its residing OCR in Rod. EPAS1 is highly expressed in Rod but lowly in Cone and BC. This OCR is a predicted LCRE of WWC1."

Figure 5b figure panels (reviewer's interpretations):

boxplots of expression of WWC1 expression in cells with different nucleotide sequences of rs6859300 in Rods, cones, and bipolar cells

EPAS1 position weight matrix aligned to reference and alternative allele sequences at rs6859300

violin plots of cellular expression of EPAS1

Genome track of WWC1 locus showing cell type specific chromatin accessibility and positioning of rs6859300 within called scATAC peaks

Example 2:

Figure 3b legend - "sc-eQTLs of the same sc-eGene are often different across cell types"

Figure 3b panel (reviewer's interpretations):

Bar graph showing proportion of eQTLs that associated with genes (sc-eGenes) expressed in one of more cell types, colored by the number of cell types that the sc-eQTL is significantly associated with.

Additionally, few of the boxplots or violin plots have descriptions of error bars (are these error bars), descriptions of median or mean, or descriptions of the 'n' and statistical test used in the analysis.

Minor points:

1. Line 124-125 "Lower correlations are observed in other cell types, particularly rare cell types, for example, a Pearson correlation of 0.41 for RGC. RGCs are not rare populations in the macula, where some of these samples were taken.
2. Line 128 - Supplementary Table 2 should be Supplementary table 3
3. Unclear which samples/cells are from macula and which are from periphery - Table 1 should be updated to include whether macula or periphery was used, including number of sequencing samples, and total number of cells.
4. Supplemental Table 2 is not useful, organize the table as sample x cell type
5. Line 143-145 - Histone marks - unclear as to where this data comes from as this is not referenced or included in any part of the Methods. If this comes from ENCODE, this is not retina tissue, so the significance of these correlations is questionable. Many of these histone marks were examined within the retina (Al Dirir et al, 2017) including annotation of cis-regulatory elements and intersection with cCRE registries
6. a. Use of Wilcoxon Rank sum across most analyses - not sure if this is appropriate in all contexts used. Additionally, significance of Wilcoxon Rank Sum p-values is questionable in figures like 2d-2e and 3f (Bipolar cells versus Rods) as the plots look essentially identical.
7. Figure 7 - Concerned that Muller glia and astrocytes show 'significance' for all diseases. This is especially important when tissues affected are not included in the dataset (choroid/sclera, lens, cornea for diseases including myopia, refractive error, and disorders of the choroid).

Dear Editor and Reviewers,

We are submitting the revision of our manuscript entitled “Single-cell multiomics of the human retina reveals hierarchical transcription factor collaboration in mediating cell type-specific effects of genetic variants on gene regulation” (GBIO-D-22-01656). We are grateful for your valuable time and effort in reviewing our manuscript. We greatly appreciate the insightful and constructive comments from reviewers for the improvement of our manuscript. Taking all comments into careful consideration, we conducted additional experiments and analyses, and revised the manuscript. We provided the responses to the comments point by point in the following sections. We hope that this revision is now suitable for publication in Genome Biology.

Thank you for your consideration and we are looking forward to your positive response.

Sincerely,

Rui Chen, PhD
Professor of Molecular and Human Genetics
Baylor College of Medicine

Kindly note that due to the presence of tracked changes, the line numbers in the main text may not be continuous. Therefore, when referring to specific line numbers, the reviewers may need to consider the nearby lines for accurate reference and context.

Editor Comments:

As you will see from the reports, the referees find the manuscript of potential interest, but they raise serious concerns that additional details and analyses are needed. In particular, Referee 1 has questions about the QTL analyses. Referee 2 requests additional support for the model and also has some concerns about the lack of detection of retinal ganglion cells. Referee 2 also suggests some additional statistical testing. It seems to us to be essential that all of the referees' concerns are fully addressed, in the form of a revised manuscript, before we can reach a final decision on publication. We will only send the manuscript back out for re-review if we feel that the concerns highlighted above have been sufficiently addressed.

Response

Thanks Editor for the comments. We have carefully considered all comments of the referees, conducted additional experiments and analyses, and revised the manuscript. We provided the responses to the comments point by point in the following sections. In response to Referee 1's comments, we conducted QTL analyses with a variety of parameters/settings, which yielded consistent conclusions that align with our initial findings. To address the concerns raised by Referee 2, we performed high-throughput reporter assays to validate the cis-regulatory elements and integrated the result with bulk ChIP-seq data of transcription factors and histone modifications from the human retina. A section titled “Massive Parallel Reporter Assays of the cis regulatory elements harboring sc-eQTLs” along with a corresponding figure panel were added to the manuscript. This integration of experimental evidence provides further support for our proposed model. Furthermore, we have provided a detailed explanation regarding the impact of low capture rate of retinal ganglion cells, as well as clarified the interpretation of the

statistical testing mentioned by Referee 2. With these additional analyses, experimental evidence, and clarifications, we believe that this revised version of the manuscript is now suitable for publication in Genome Biology.

Reviewer reports:

Reviewer #1: Wang et al provided a comprehensive single-cell multiomic analysis of major cell types of the human retina. They mapped eQTL and caQTLs which were well supported by allele-specific expression and chromatin accessibility, and discovered new QTLs missed by earlier studies using bulk sequencing. Importantly, they observed many cell type-specific genetic regulatory effects and presented thought-provoking analyses implicating a sophisticated transcription factor network underlying cell type specificity. Lastly, they performed a well-designed fine-mapping analysis to identify potential causal variants and relevant cell types. Even though some details were currently missing, the statistical analyses were appropriate and supported the authors conclusions. This work is highly relevant and very appropriate for Genome Biology. I have the following comments and questions for the authors.

Response:

We are grateful for the encouraging and positive comments by Reviewer #1.

- The authors omitted some important details on the identification of QTL in their data. In particular, why did they choose to include N = 3 PEER factors and 1 genotype PC? Did top PEER factors correspond to known technical covariates? What would happen to the power to detect eQTL if more PEER factors were included? How about other demographic variables such as sex and age?

Response:

Thanks Reviewer #1 for the constructive comments. We added the following clarification to the section "The selection of covariates for eQTL mapping" in the Supplementary Information page 14.

"Given our relatively small cohort of N=20, we made a careful decision to include N=3 PEER factors and 1 genotype principal component (PC) as covariates. This choice was made to ensure sufficient statistical power, account for the systematic confounding effects in gene expression with the PEER factors, and prevent overfitting through enabling accurate estimation of each variable in the linear regression (specifically, each variable in the linear regression can be estimated with $20/5=4$ subjects)¹.

The observed correlations between the top PEER factors and the quality control metrics of snRNA-seq suggested that these PEER factors correspond to known technical covariates. Specifically, in the case of rod cells, the 1st PEER factor is positively correlated with percent of reads mapped antisense to gene (Pearson correlation $r=0.51$, $p=0.02$); the 2nd PEER factor is positively correlated to median UMI counts per cell ($r=0.47$, $p=0.04$); and the 3rd PEER factor is positively correlated to fraction of reads in cells ($r=0.77$, $p=7.14e-05$).

To assess the impact of including additional PEER factors in eQTL analysis, we conducted eQTL analysis including N=7 PEER factors and 1 genotype principal component (PC) as covariates. Surprisingly, we observed a significant reduction in detection power, as the number

of gene-level significant eQTLs decreased from 14,377 to 3,714. This finding suggests that including a higher number of PEER factors for a small sample cohort might introduce excessive correction and inadvertently suppress true eQTL signals. Furthermore, we performed another eQTL analysis incorporating 3 PEER factors, 1 genotype PC, age, and sex as covariates. Interestingly, this analysis also resulted in a reduced power to detect eQTLs, with the number of gene-level significant eQTLs decreasing from 14,377 to 12,567. Notably, we observed a negative correlation between the 3rd PEER factor and the age of the samples (Pearson correlation $r=-0.50$, $p=0.02$). This finding suggests that the 3rd PEER factor partially captures the age effect in the data.

Considering these results, we have made the decision to utilize the current set of covariates, which includes 3 PEER factors and 1 genotype PC. This choice balances the need for confounding correction while preserving the detection power for eQTL analysis."

- In addition, for caQTL mapping, what is the window for SNPs to be considered for each peak? Did the authors distinguish caQTLs where the SNP overlap with the peak, and those that are outside? (Maybe that's the "distal caQTLs" they mentioned in line 248, page 10, but the description of this distinction was not easily found).

Response:

Thanks Reviewer #1 for the question. Given the small cohort of $N=20$, to mitigate the reduction in caQTL detection power caused by multiple testing correction, we focused the caQTL detection on the SNPs located within each peak. The rationale is that the snATAC-peaks are enriched with potential cis regulatory elements, and SNPs residing within a peak are most likely to impact the accessibility of the corresponding peak². Specifically, we calculated the association between each SNP within a peak and the corresponding peak. We mentioned that in the section "Significant proportion of sc-caQTLs are cell type specific in retina" Page 8 line 456-457.

We apologized for any confusion regarding the term "distal caQTLs". We referred this type of caQTLs to the sc-caQTL identified by the above approach but located in the non-promoter snATAC-peaks. To clarify, we revised the sentence as "Notably, for sc-caQTLs located in non-promoter OCRs, those shared among multiple cell types displayed significantly greater effect than those unique to one cell type (e.g., Rod, one-sided Wilcoxon rank sum test, $p < 2.2 \times 10^{-16}$, Fig. 4d, Supplementary Table 9)." Page 8 line 468-498.

However, in the later section "Interaction among OCRs", for each common variant within snATAC-peaks, we tested the association between the variant and the accessibility of snATAC-peaks in $-/+250\text{kb}$ surrounding the variant. The detailed can be found in the Methods section Page 25 line 1403-1405.

- The authors observed that variants in wider OCRs have local effects, while the variants in narrow OCRs are more likely to affect accessibility of the entire OCR. This is interesting, although a confounding observation can be that the wider peaks called are less accurate (may be a union of multiple smaller peaks). What is the distribution of peak lengths in this dataset? An alternative approach for caQTL calling is to use the "tiled" peaks (defined as 500bp windows genome-wide in ArchR) and then map significant caQTLs to OCRs. Have the authors tried this

approach of mapping caQTLs?

Response:

Thanks Reviewer #1 for the constructive comments. The medians of peak lengths are around 1kb. The distribution of peak lengths was shown in the figure below.

To address the potential confounding effect caused by wider peaks that may be called less accurately, we revised the analysis by considering only the peaks with a length of less than 2kb. This selection criterion ensures that we focus on narrower, more precise peaks that are less likely to encompass multiple distinct regulatory regions. As a result, we still observed that variants in wider OCRs exerted local effects, while the variants in narrow OCRs were more likely to impact the accessibility of the entire OCR (Supplementary Figure 3d, Rod, p-value=7.7e-13).

Additionally, an independent study, utilizing bulk ATAC-seq on cell line and applying different approaches for peak calling, caQTL and ASCA analyses compared to ours, arrived at the same conclusion as our study, providing further evidence to support this result (Extended Data Fig. 6a. in their paper)².

Furthermore, in our preliminary analysis, we explored the approach of conducting caQTL analysis based on the peaks called by MACS2 through ArchR with a fixed length of 501bp for all

peaks. The number of caQTLs detected per cell type is comparable to the numbers identified by our current approach. Given the variation in the length of ATAC-seq peaks, which can reflect biological phenomena such as transcription factor collaboration (e.g., multiple TFs binding to an enhancer), we utilized our current approach in our study, which allows variable peak length.

- The authors made the interesting observation that some cell-type specific caQTLs are associated with caPeaks which are accessible across multiple cell types. How did the authors address the possibility that there may be cases where the SNP is actually a caQTL in multiple cell types but did not pass statistical threshold because of differences in power, as a result of cell type abundance differences?

Response:

Thanks Reviewer #1 for the constructive comment. To assess the impact of cell type abundance differences on the caQTL analysis, we randomly sampled 400 cells per cell type per individual to repeat the caQTL analysis. As a result, we found the consistent trends that aligned with our initial findings based on the original dataset. We added the following section “Assessing the potential bias in caQTL detection introduced by cell type abundance differences by down-sampling” in Supplementary Information Page 15 line 287-303.

“To assess the impact of cell type abundance on sequencing depth (in cell type level) and thereby the number of sc-caQTLs detected per cell type, we performed a random down-sampling approach. We randomly sampled 400 cells per cell type per individual from the original dataset, allowing for sampling with replacement. Through this analysis, we consistently observed trends that aligned with our previous findings based on the original dataset. A total of 2,840 sc-caQTLs were detected, ranging from 98 to 814 per cell type. Notably, the majority of these sc-caQTLs were cell type-specific, with 74.5% to 90.1% being unique to a single cell type (Supplementary Fig. 4a, b). Furthermore, we still found that the majority (85.7%) of the cell type specific sc-caQTLs were associated with caPeaks (namely, the residing peaks of caQTLs) that are accessible across multiple cell types (Supplementary Fig. 4c). Overall, by conducting this random sampling analysis, we have addressed the potential bias introduced by cell type abundance differences, and our results consistently validated the patterns observed in the original dataset.”

- The calculation of LD was not clearly presented. Given that this cohort has individuals from diverse ancestries, which reference panel was used to assess LD?

Response:

The LD was estimated based on the 20-individual dataset using PLINK. Please see “Identification of LD-independent sc-caQTL and LD-independent sc-eQTL” in the Methods section for details page 25 line 1392-1399.

- In Figure 3F, the authors noted that the expression levels of TFs disrupted by eQTLs in the relevant cell types have higher expression than non-eQTL cell types. This is a very interesting observation and a convincing analysis to justify their arguments that trans factors may drive cell type-specific caQTLs. How many TFs were included in this analysis (predicted to have motifs perturbed?) Is there evidence from published studies supporting some of the TFs in establishing cell type specificity?

Response:

2817 TF motifs (Hsapiens) in the MotifBreakR database were considered in the analysis of predicting the motif disrupting effects of SNPs. In Figure 3F, 95-98 of the 2817 TFs were

predicted to have motifs perturbed by eQTLs and have gene expression ≥ 50 CPM in rod cells (Supplementary Table 7). Please see “Predicting the motif disrupting effects of SNPs” in the Methods section for details (page 25 line 1372-1390). Furthermore, we revised Figure 3f and 5b to only consider the unique transcription factors (previously we counted a TF multiple times, if its motif was perturbed by multiple variants in different genomic regions respectively) to avoid potential inflation, and labeled the number of TFs in the figures.

Among these TFs with motifs disrupted by eQTLs or caQTLs, some of the TFs were known in establishing cell type specificity for retinal cells, for example, OTX2, CRX, NRL, and MEF2D in Rod and Cone; OTX2 and VSX2 in BC; ONECUT1 and ONECUT2 in HC; TFAP2A and TFAP2B in AC; LHX2 and NFIX in MG³⁻⁹.

- The enrichment test of caQTLs associated with dependent OCRs in Figure 6A is interesting. How were the background "all tested variants" defined? It seems that the authors identified variants which are not caQTLs in the master peak but still are associated with distal OCRs (e.g. ~10% in Rod). Could the authors provide insights on this class of QTLs?

Response:

The “all tested variants” are all the variants tested for caQTLs, which are the variants in OCRs (snATAC-seq peak) with $AF \geq 0.1$. Two types of hypotheses may explain the variants that are not caQTLs in the master peak but associated with distal OCRs: 1) This phenomenon may be caused by certain biological mechanism. For example, these variants may disrupt the binding of a TF in the master peak but not be able to change the chromatin accessibility of the entire master peak due to the binding of other TFs in the same peak. However, the disruption of the TF binding may cause the change of chromatin accessibility of distal OCRs due to the interaction between OCRs. 2) This phenomenon may represent the false positives that the variants are associated with distal OCRs by chance but not biologically meaningful. Therefore, they can be used to estimate background noise and should be considered when determining the actual proportion of variants that lead to chromatin change of both the master and distal OCRs through a biological mechanism. Since the 1st hypothesis remains unexplored, the 2nd hypothesis may be a more suitable explanation for our current analysis.

- In Figure 6C, please indicate what statistical tests were performed (compared to all tested OCRs or comparing between master and dependent OCRs). Similarly, the p-values should be reported in the text when comparing master and dependent OCRs in active chromatin (H3K27ac and H3K4me2).

Response:

Thanks Reviewer #1 for the constructive comment. We added the following description to the figure legend of Figure 6C: “Two-sided Binomial test was performed to compare the proportion of master_OCR or dependent_OCR with the proportion of tested_OCR.”

Additionally, we added the p-values in the main text for comparing master and dependent OCRs in DAR and active chromatin: “Furthermore, we observed a slightly higher enrichment in DARs in the master regions compared to the dependent regions (one-sided Fisher’s Exact test, caPeak: $p = 7.1 \times 10^{-3}$; caPeaks associated with LCREs: $p = 0.082$), as well as an enrichment

in active enhancer epigenetic modifications (the concurrent H3K27ac and H3K4me2, one-sided Fisher's Exact test, caPeak: $p = 1.48 \times 10^{-10}$; caPeaks associated with LCREs: $p = 0.10$) (Fig. 6c)." Page 11 line 646-651.

- As a related question, could the authors provide more information on the relationship between master and dependent OCRs? Are they likely co-accessible peaks from scATAC, or is there relevant Hi-C data supporting their interactions?

Response:

Thanks Reviewer #1 for the great suggestion. We added the following analysis to the main text: "Furthermore, a significant proportion (59.7%) of the master OCRs are co-accessible with at least one of the corresponding dependent OCRs (correlation ≥ 0.2 , FDR < 0.05), showing a 2.8-fold enrichment compared to the background (the co-accessibility of two random peaks within 250kb region, two-sided binomial test $p = 0$), further supporting the interactions between master and dependent OCRs." Page 11, line 639-643.

- It is very interesting to compare the caQTLs affecting multiple peaks in concordant and opposite directions, and the authors presented convincing data supporting compensatory roles of OCRs in leading to cis-eQTLs. Again, is there evidence (from co-accessibility or Hi-C, for example) supporting their interactions in case they jointly determine a cis-eQTL?

Response:

Thanks Reviewer #1 for the constructive comment. We added the following analysis to the main text: "In addition, the majority (71.4%) of sc-caQTLs that affect multiple peaks and also overlap with cis-eQTLs exhibit co-accessibility of the affected peaks, showing 1.23-fold enrichment compared to sc-caQTLs that affect multiple peaks but do not overlap with cis-eQTLs (two-sided binomial test, $p = 1.96 \times 10^{-5}$). This result provides further evidence supporting the interaction of multiple peaks may jointly determine a cis-eQTL" Page 12 line 708-712.

- Among a total of 818 fine-mapped variants from three GWAS, the authors identified 27 QTLs in their dataset. Is this fraction higher than expected? (What fraction of GWAS SNPs were tested for QTLs in this study?) Further, could the authors provide a comparison of fine-mapping using existing data from bulk eQTL and annotated cis-regulatory regions, and demonstrate the gain in power to nominate potential causal variants in these traits, using single-cell multiomics and single-cell QTL mapping?

Response:

Thanks Reviewer #1 for the constructive comment. Among a total of 9762 SNPs tested for GWAS, 1130 SNPs were also tested for single cell QTLs/ASCAs in our study, and 224 (19.8%) of the 1130 SNPs were overlapped with QTLs/ASCAs. In contrast, among a total of 816 fine-mapped GWAS SNPs (we modified the fine-mapping analysis, so there is minor difference in the numbers between two versions, e.g. 816 vs. 818), 261 SNPs were tested for single cell QTLs/ASCAs in our study, and 38 (14.6%) of the 261 SNPs were overlapped with single cell QTLs/ASCAs. Therefore, we did not observe a higher fraction of the fine-mapped GWAS SNPs

overlapped with QTLs/ASCAs (14.6%) than expected (19.8%) (two-sided Fisher's Exact test, $p = 0.053$). The potential reasons may be: 1) the small cohort of $N=20$ may result in reduced power for detecting QTL. 2) The examined GWAS loci might have higher overlapping rate with the QTLs from other tissue or cell types outside of retina. Specifically, age-related degeneration GWAS loci may be enriched in retinal pigment epithelium^{10,11}, myopia/fraction error¹² and open-angle glaucoma loci may be also enriched in the anterior segment of the eye¹³.

Following Reviewer #1's suggestion, we performed the fine-mapping of GWAS variants using the annotated cis-regulatory regions from the ENCODE cCRE registry ($N= 959449$ regions)¹⁴ and identified 841 GWAS variants with $PIP > 0.1$, which is comparable to the number based on our single-cell multiomics study ($N=816$). Out of the 841 fine-mapped GWAS variants, 826 were tested for bulk retinal eQTLs¹⁵. Among the tested variants, 216 (26.15%) overlap with bulk retinal eQTLs, showing **1.79-fold** higher mapping rate compared to the single cell QTLs/ASCAs (two-sided Fisher's Exact test, $p = 7.7 \times 10^{-5}$). Considering that the bulk retinal eQTL study was conducted with a cohort of **22.65-fold** more individuals ($N=453$) compared to our single cell association study ($N=20$), our single-cell multiomics and single-cell QTL mapping methods exhibit increased power in nominating potential causal variants for these GWAS variants. Furthermore, since gene regulation is highly cell type/context dependent, our single cell study provides a higher resolution in identifying the relevant cell types where these potential causal variants exert their effects.

Given the above analyses, we changed the sentence "Moreover, we showed that integration of single cell multiomics and GWAS studies can increase **the power to prioritize** effective cell context, causal variants and genes, and better dissect the underlying regulatory mechanisms." to "Moreover, we showed that integration of single cell multiomics and GWAS studies can increase **the resolution to nominate** effective cell context, causal variants and genes, and better dissect the underlying regulatory mechanisms". Page 16 line 1017-1019.

- The authors may consider clarifying the concept of "hierarchical transcription factor" network they proposed in this work, as it is not obvious what they refer to. Did they want to highlight the potential role of pioneer transcription factors, which establish chromatin accessibility and recruit other cell-type specific factors?

Response:

Thanks Reviewer #1 for the constructive comment. Indeed, for "hierarchical transcription factor collaboration", we referred to the process by which the pioneer transcription factors first establish chromatin accessibility and then recruit other cell-type specific factors.

We add a description in the Discussion section to clarify the concept of "hierarchical transcription factor" network page 16 line 1003-1008.

"Specifically, we hypothesized hierarchical transcription factor collaboration may be crucial in gene regulation, whereby the pioneer factors first establish chromatin accessibility of the regulatory genomic regions in cells sharing a similar lineage, resulting in different cell types sharing common OCRs, then additional cell type specific factors may be recruited to these OCRs later in a collaborative manner to regulate gene expression".

- I would suggest that the authors make changes to some figure legends to describe the data

rather than interpret them. For example, in Figure 2g, what are the columns and rows of the heatmap? As another example, in Figure 4f-g, please describe the axes in figure legends and move the interpretation to the main text.

Response:

We greatly appreciate the suggestion of Reviewer #1. We revised the figure legends and made them as a description of the graphs and the presented data.

- The authors should consider using language more consistent with the current framework for describing population characteristics; for example, the use of the term "Caucasian" should be carefully avoided based on recommendations by the National Academies (https://urldefense.proofpoint.com/v2/url?u=https-3A__nap.nationalacademies.org_catalog_26902_using-2Dpopulation-2Ddescriptors-2Din-2Dgenetics-2Dand-2Dgenomics-2Dresearch-2Da-2Dnew&d=DwIGaQ&c=ZQs-KZ8oxEw0p81sqgiaRA&r=dcxpvJ-E4MMZd6unJFaTyA&m=IDleE0cDRS4IJ-a7hAqJ5AvSuTK2HbYGqTrCzKpIIghiChDINxuQ7sOaZ63NwsdZ&s=2W27wIPdLpcRnJmleij_QhgF9NdKsT9ia920Q0VWvWQ&e=)

Response:

Thanks Reviewer #1 for the suggestion. We changed the term "Caucasian" to "European", and the term "Hispanic" to "Latino".

- Some references were not not correctly labeled. For example, in page 14, the authors wrote that "We incorporated functional annotation (including OCR and LCRE derived from single cell multiomics data) of variants to prioritize GWAS loci[48]56" but neither of the references matched their described methods.

Response:

Thanks Reviewer #1 for the constructive comment. We double checked the reference. We deleted the reference 56 which was for an earlier version of the manuscript.

We followed the fine-mapping approach in the reference [48]: "Transcriptome and regulatory maps of decidua-derived stromal cells inform gene discovery in preterm birth", which were described in the "Integrated analysis of GWAS and decidual cell functional annotations improves fine mapping of causal variants of gestational duration and identifies putative target genes" in the results section and "Fine-mapping GWAS loci associated with gestational length" in the methods section of the paper.

Reviewer #2: In the manuscript by Wang et al., the Authors utilize single-cell RNA-sequencing, separate single-cell ATAC-sequencing, and whole genome sequencing of 20 healthy patient retinas to examine how genetic sequence variability affects chromatin structure and gene transcription within individual retinal cells. This large body of work provides a significant dataset where precise genetic information can be correlated with epigenetic and transcriptomic features

within the mature retina, providing hypothesis-driving insights for mechanisms by which genetic variability results in phenotypic consequences.

Response:

We greatly appreciate the positive comments and recognition of our work by Reviewer #2.

The manuscript presents a heroic integration of a mountain of data, but suffers from major deficiencies.

1) The manuscript in its current form is an accumulation of correlated results between genetics, chromatin accessibility, and transcription, resulting in a broad survey of results listed as '% X of Y shows enrichment of Z'. The Authors do show specific examples of results (Figure 3d-e; 5a-b; 6d-7d) but the biological significance of any of these findings is not examined. The model (Figure 8) is built on the assumptions that transcription factor affinity is changed. While this may in fact be the correct conclusion, any reference to binding affinities of transcription factors is inferred based on canonical motifs, but not actual data.

Response:

Thanks Reviewer #2 for the constructive comment. To support our findings, we performed additional high-throughput reporter assay to assess the regulatory activities of the open chromatin sequences associated with the identified sc-eQTLs, and integrated the results with the public ChIP-seq data of transcription factors and histone modifications from the adult human retina.

We added the following section "Massive Parallel Reporter Assays of the cis regulatory elements harboring sc-eQTLs" in the manuscript page 12 line 714-760:

"To validate the results of our association studies, we conducted Massive Parallel Reporter Assays (MPRAs) to assess the regulatory activities of 931 open chromatin sequences associated with the index sc-eQTLs identified in human rod cells¹⁶. These sequences were tested in explanted mouse retinas during postnatal day 0 to 8, which primarily consist of rod cells (Fig.7a). The MPRA library was designed with synthesized 224bp oligonucleotides (oligos) centered on the peak summit of the 931 open chromatin regions, which were identified through snATAC-seq and associated with the index sc-eQTLs, along with positive and negative control sequences (Methods). Due to the relatively low conservation in noncoding nucleotides between human and mouse genomes, we only included the sequences containing the human genome reference nucleotide (hg19) for the MPRA. Following the MPRA, we calculated an activity score for each library sequence and normalized it based on the activity of the basal Crx promoter. As a result, we identified 258 enhancers and 66 silencers that showed at least a twofold higher or lower activity than the basal Crx promoter (q-value < 0.05), as well as 607 inactive sequences that exhibited activity within a twofold change of the basal activity or not significant different from the basal activity (q-value \geq 0.05) (Fig. 7b,c, Supplementary Table 1). A high fraction of sequences identified as inactive could be attributed to several factors: 1) limited sensitivity of the experimental system; 2) differential regulatory activity of CREs between human and mouse; and 3) false positives in identified sc-eQTLs due to a small cohort.

By integrating the MPRA result with bulk ChIP-seq data from the adult human retina, we observed that the validated enhancers are significantly enriched of the binding of photoreceptor-specific transcription factors, such as OTX2 (68.2% vs. 41.4%, one-sided Fisher's exact test,

$p = 2.7 \times 10^{-13}$), CRX (35.7% vs. 15.7%, one-sided Fisher's exact test, $p = 1.8 \times 10^{-10}$) and MEF2D (28.7% vs. 12.7%, one-sided Fisher's exact test, $p = 3.3 \times 10^{-8}$), compared to the inactive sequences (Fig. 7d). Additionally, these enhancers exhibit a higher presence of active enhancer epigenetic markers, the concurrent H3K27ac and H3K4me2, than the inactive sequences (76.4% vs. 59.5%, one-sided Fisher's exact test, $p = 9.7 \times 10^{-7}$, Fig. 7d). In contrast, the validated silencers are enriched of CTCF binding, compared to the inactive sequences (one-sided binomial test, 71.2% vs. 55.5%, one-sided Fisher's exact test, $p = 9.4 \times 10^{-3}$, Fig. 7d). These results provide experimental evidence to support that a significant proportion of OCRs harboring sc-eQTLs are indeed active cis regulatory elements, suggesting that perturbation of these elements by genetic variants may lead to changes in gene expression. Interestingly, the validated enhancers are enriched of the co-binding of an early lineage determining factor, OTX2, with at least one of cell type/context specific factors, CRX, NEF2D and NRL, compared to the inactive elements (62.5% vs. 46.2%, one-sided Fisher's exact test, $p = 6.2 \times 10^{-4}$) (Fig. 7e). This result suggests that the collaboration between early lineage determining factors and cell type/context specific factors may be crucial for gene regulation, supporting our findings.”

Fig. 7

2) The use of acronyms throughout the manuscript makes the dialog almost impossible to follow. This reviewer struggled to read most of the document, needing a reference table of acronyms to try and decipher what the Authors were trying to analyze and resulting conclusions

from the analyses.

Response:

Thanks Reviewer #2 for the constructive comment. We apologized for the inconvenience in reading the manuscript, and added a acronyms list in the Supplementary Information Page 16.

3) Concerns about the significance of analyses utilizing snRNA-seq with snATAC-seq
a. Ideally this would have been done in the SAME cells using the multiome kit so that chromatin accessibility, and allelic expression could be matched within individual cells. It is recognized that these experiments, however, were started prior to the availability of the multiome kit. Authors should state the limitations of the analysis because of this, especially when differences major differences in cell type proportions captured from individual patients (see point b) are significant.

Response:

Thanks Reviewer #2 for the constructive comment. We appreciated the understanding of Reviewer #2 that our single cell sequencing of gene expression and chromatin accessibility were performed prior to the availability of the multiome kit. However, not measuring gene expression and chromatin accessibility in the same cell simultaneously may have minor impact on our current study, because all our genetic association studies (i.e., eQTL, caQTL, ASE, ASCA) were performed in cell type level not individual cell level. Specifically, gene expression or chromatin accessibility was measured in the pseudo bulk of cells from the same cell type. Furthermore, both the snRNA-seq cells and snATAC-seq cells are from the same cohort of individuals.

Following Reviewer #2's suggestion, we added the following sentences to the manuscript:

“Furthermore, differences in cell type proportions can also impact the detection power of genetic association studies. This is because accurate measurement of chromatin accessibility and allelic-specific effects relies on sufficient sequencing depth, which can be influenced by differences in cell type proportions.” Page 17 line 1068-1071.

“Another potential improvement is the utilization of multiome kits, enabling simultaneous profiling of chromatin accessibility and gene expression within the same cell. This would allow for the matching of chromatin accessibility and allelic expression within individual cells, thereby increasing the power to gain insights into gene regulation.” Page 17 line 1078-1081.

Indeed, the differences in cell type proportions may have impact on comparing gene expression or chromatin accessibility metrics across different cell types, especially when mapping chromatin accessibility QTL or allelic specific chromatin accessibility (ASCA). To address this concern, we down-sampled 400 snATAC-seq cells per cell type per individual and repeated caQTL analysis. As a result, we found consistent trends that aligned with our initial findings based on the original dataset. Please find the details in the “Assessing the potential bias in caQTL detection introduced by cell type abundance differences by down-sampling” section in the Supplementary Information Page 15 line 287-302.

b. Throughout the manuscript, the authors examine cell-type specific patterns of

expression/accessibility, but fail to include the retinal ganglion cells (RGCs). This is likely because capture rate of the RGCs is extremely low in the snATAC (Figure 1). All conclusions of cell-type specificity are therefore limited because a major and disease-relevant cell type is not included. There is no mention of the limitations and/or biases of the analysis when missing this cell type. Conversely, when the Authors examine relevant diseases (Figure 7), the RGCs are now included because primary open angle glaucoma is a disease where RGCs are affected. As the dataset is likely under-powered for RGC analyses, it seems disingenuous to state that RGCs don't have genetic features (accessibility/expression) that are likely to contribute to POAG.

Response:

Thanks Reviewer #2 for the constructive comment. Indeed, the capture rate of RGCs per individual is very low for the snATAC-seq, therefore, we did not include RGC in genetic association studies that required individual (donor) level data, such as QTL and ASCA. However, when examining cell type enrichment of GWAS loci (Figure 8), we included RGC for evaluation based on both chromatin accessibility (derived from snATAC-seq) and gene expression (derived from snRNA-seq). Because the cell type enrichment analysis is based on snATAC-peaks or gene expression derived from RGC cells aggregated across all 20 donors, rather than per donor basis. In total, we obtained 7,789 RGCs for snRNA-seq and 2,442 RGCs for snATAC-seq, as well as 109,398 snATAC-peaks from RGCs. Importantly, the number of snATAC-peaks (n=109,398) identified in RGCs falls within a similar range as the number of snATAC-peaks (n=124,653) identified in Rods. Therefore, in cell type level (aggregated across 20 donors), we should have adequate power to analyze the enrichment of GWAS loci in RGC. Additionally, an independent study also found similar results as ours, namely, Muller glia and astrocyte rather than RGC are the retinal cell types that are enriched of primary open-angle glaucoma (POAG) and elevated intraocular pressure (IOP) GWAS loci¹³.

Following Reviewer's suggestion, we added the following sentences to the discussion section in the manuscript: "Additionally, the low capture rate of RGCs and astrocytes per individual due to cell type proportion differences limited their inclusion in the genetic association studies. This compromised our ability to effectively fine-map primary open-angle glaucoma (POAG) by overlapping with sc-QTLs, as RGCs are the affected cell type in glaucoma. We anticipate that enriching for cell types representing a small proportion of the total retinal cell population will increase the power to map QTLs, identify allele-specific variants, and nominate causal variants and genes for GWAS loci." Page 17 line 1071-1078.

c. snRNA-seq seems to show multiple rod populations, including a mixing of rods with many other cell types (bipolars, Mullers, others?). Potential causes of this variability is high amounts of ambient RNA (concern of snRNA-seq) and/or doublets. The methods state that 'SoupX' and 'DoubletFinder' were applied to remove doublets, but a low doublet threshold was utilized (0.01). The standard doublet rate for capturing 10,000 cells/run on a 10x Genomics v3.1 3' is 8% (0.08 doublet threshold).

Response:

Thanks Reviewer #2 for the question. We apologize for any confusion caused regarding the doublet rate threshold. We would like to clarify that we used "*the cell number*/1000 × 0.01", rather than 0.01, to estimate doublet rate (Page 21 line 1227). Thus, for capturing 10,000

cells/run on a 10x Genomics v3.1 3', the doublet rate threshold we applied is $\frac{10000}{1000} \times 0.01 = 10\%$, which is higher than 8%. Since our analyses were primarily conducted in cell type level rather than individual cells through aggregating cells of the same cell type into a pseudo-bulk, we anticipate that a small proportion of potential doublets present in the data would have a minimal impact on our results and overall patterns.

d. Would be helpful to have UMAPs of scRNA and scATAC colored by sample to observe potential batch effects

Response:

Thanks Reviewer #2 for the constructive comment. We have plot UMAPs of snRNA cells and snATAC cells colored by sample/batch source in the Supplementary Fig. 1a and 1b respectively. These figures showed that batch effects within our data are minimal.

Supplementary Fig. 1a and 1b

4) Figure Legends - Most figure legends are interpretations of the data, not an actual description of the graphs, presented data. Here are two examples, but this is how the legends of a MAJORITY of the figures read.

Example 1:

Figure 5b legend (Figure 5c is also labeled as 5b) - "The variant rs6859300 is a Rod-specific sc-caWTL of its residing OCR, and resides in an OCR accessible in ROD, Cone, and BC. This variant is predicted to enhance binding of EPAs1, increasing chromatin accessibility of its residing OCR in Rod. EPAS1 is highly expressed in Rod buy lowly in Cone and BC. This OCR is a predicted LCRE of WWC1."

Figure 5b figure panels (reviewer's interpretations):

boxplots of expression of WWC1 expression in cells with different nucleotide sequences of rs6859300 in Rods, cones, and bipolar cells

EPAS1 position weight matrix aligned to reference and alternative allele sequences at rs6859300

violin plots of cellular expression of EPAS1

Genome track of WWV1 locus showing cell type specific chromatin accessibility and positioning of rs6859300 within called scATAC peaks

Example 2:

Figure 3b legend - "sc-eQTLs of the same sc-eGene are often different across cell types"

Figure 3b panel (reviewer's interpretations):

Bar graph showing proportion of eQTLs that associated with genes (sc-eGenes) expressed in one of more cell types, colored by the number of cell types that the sc-eQTL is significantly associated with.

Additionally, few of the boxplots or violin plots have descriptions of error bars (are these error bars), descriptions of median or mean, or descriptions of the 'n' and statistical test used in the analysis.

Response:

We greatly appreciated Reviewer #2 for the detail suggestions. We revised the figure legends according to the suggestions.

Minor points:

1. Line 124-125 "Lower correlations are observed in other cell types, particularly rare cell types, for example, a Pearson correlation of 0.41 for RGC. RGCs are not rare populations in the macula, where some of these samples were taken.

Response:

Thanks Reviewer #2 for the constructive comment. We revised the sentence to "Lower correlations are observed in other cell types, particularly the cell types that represent a small proportion of the total retinal cell population, for example, a Pearson correlation of 0.41 for RGC". Page 5 line 257-259.

2. Line 128 - Supplementary Table 2 should be Supplementary table 3

Response:

Thanks Reviewer #2 for the constructive comment. We changed Supplementary Table 2 to Supplementary table 3.

3. Unclear which samples/cells are from macula and which are from periphery - Table 1 should be updated to include whether macula or periphery was used, including number of sequencing samples, and total number of cells.

Response:

Thanks Reviewer #2 for the suggestion. We have updated the Supplementary Table 1 accordingly.

4. Supplemental Table 2 is not useful, organize the table as sample x cell type

Response:

Thanks Reviewer #2 for the suggestion. We have organized the Supplementary Table 2 accordingly.

5. Line 143-145 - Histone marks - unclear as to where this data comes from as this is not referenced or included in any part of the Methods. If this comes from ENCODE, this is not retina tissue, so the significance of these correlations is questionable. Many of these histone marks were examined within the retina (Al Diri et al, 2017) including annotation of cis-regulatory elements and intersection with cCRE registries

Response:

Thanks Reviewer #2 for the question. The H3K27ac and H3K4me2 bulk ChIP-seq data we utilized in the manuscript were obtained from the adult human retinal tissue published previously¹⁷. The detailed information about data source and processing can be found in the section "Processing of the bulk ATAC-seq and ChIP-seq data" in Supplementary Information Page 13-14 line 242-251.

6. a. Use of Wilcoxon Rank sum across most analyses - not sure if this is appropriate in all contexts used. Additionally, significance of Wilcoxon Rank Sum p-values is questionable in figures like 2d-2e and 3f (Bipolar cells versus Rods) as the plots look essentially identical.

Response:

Thanks Reviewer #2 for the comment. We would like to clarify that we utilized various statistical tests such as the Binomial test, Fisher's Exact test, and Wilcoxon Rank sum test based on the specific statistical question or context. The Wilcoxon rank-sum test, also known as the Mann-Whitney U test, is a nonparametric statistical test used to compare whether two independent samples come from the same distribution. The Wilcoxon rank-sum test does not rely on specific distributional assumptions and is suitable when sample size is small. Given these characteristics, the Wilcoxon rank-sum test appears to be an appropriate choice for comparing two distributions in our study.

Additionally, we thoroughly reviewed and confirmed the accuracy of the p-values associated with Figures 2d, 2e and 3f. Furthermore, we revised Figure 3f and 5b to only consider the unique transcription factors (previously we counted a TF multiple times, if its motif was perturbed by multiple variants in different genomic regions respectively) to avoid potential inflation. The Figures 2d, 2e, 3f and 5b were re-plot in violin plot format for better visualization. For transparency and reproducibility purposes, we provided the original data in Supplementary Table 5, 6, and 7, enabling others to reproduce the results and obtain the corresponding p-values for Figures 2d, 2e and 3f.

Fig. 2

Fig.3f

7. Figure 7 - Concerned that Muller glia and astrocytes show 'significance' for all diseases. This is especially important when tissues affected are not included in the dataset (choroid/sclera, lens, cornea for diseases including myopia, refractive error, and disorders of the choroid).

Response:

We appreciate Reviewer #2's comment. In Figure 7, we presented the raw p-values without applying multiple testing correction. However, it is important to note that in the main text (page 13 line 764-777), we mentioned that after applying multiple testing correction with a false discovery rate ($p_{adj} < 0.1$) through Benjamini-Hochberg correction, the enrichment of GWAS loci in specific cell types is limited. Specifically, cup areas (CA) and vertical cup-disc ratio (VCDR) of optic nerve, intraocular pressure (IOP), and primary open angle glaucoma (POAG) displayed enrichment in astrocytes and MG, which was replicated in an independent study¹³.

Refractive error and myopia loci displayed enrichment in most of major retinal cell types, which was suggested by an independent study as well¹². The loci associated with choroid/retina disorders, retinal detachments/breaks, and retinal problems showed enrichment in MG. Consistently, MG has been suggested to be associated with various retinal diseases^{18,19}. While we acknowledge that our dataset for the retina does not include the tissues primarily affected by some of these eye complex diseases, our results do not negate the importance of other affected tissues being enriched for GWAS loci. Rather, our focus is on studying which specific cell types in the retina exhibit enrichment of GWAS loci. These two aspects are not mutually exclusive. It is plausible that both the cell types in the retina and the affected tissues outside the retina could show significant enrichment of the same GWAS trait¹³.

Reference

1. Austin, P. C. & Steyerberg, E. W. The number of subjects per variable required in linear regression analyses. *J Clin Epidemiol* **68**, 627–636 (2015).
2. Liang, D. *et al.* Cell-type-specific effects of genetic variation on chromatin accessibility during human neuronal differentiation. *Nat Neurosci* **24**, 941–953 (2021).
3. de Melo, J. *et al.* Lhx2 Is an Essential Factor for Retinal Gliogenesis and Notch Signaling. *J Neurosci* **36**, 2391–405 (2016).
4. Sapkota, D. *et al.* Onecut1 and Onecut2 redundantly regulate early retinal cell fates during development. *Proceedings of the National Academy of Sciences* **111**, (2014).
5. Clark, B. S. *et al.* Single-Cell RNA-Seq Analysis of Retinal Development Identifies NFI Factors as Regulating Mitotic Exit and Late-Born Cell Specification. *Neuron* **102**, 1111–1126.e5 (2019).
6. Andzelm, M. M. *et al.* MEF2D Drives Photoreceptor Development through a Genome-wide Competition for Tissue-Specific Enhancers. *Neuron* **86**, 247–263 (2015).
7. Yamamoto, H., Kon, T., Omori, Y. & Furukawa, T. Functional and Evolutionary Diversification of Otx2 and Crx in Vertebrate Retinal Photoreceptor and Bipolar Cell Development. *Cell Rep* **30**, 658–671.e5 (2020).
8. Jin, K. *et al.* Tfp2a and 2b act downstream of Ptf1a to promote amacrine cell differentiation during retinogenesis. *Mol Brain* **8**, 28 (2015).
9. Bian, F. *et al.* Functional analysis of the Vsx2 super-enhancer uncovers distinct cis-regulatory circuits controlling Vsx2 expression during retinogenesis. *Development* **149**, (2022).
10. Winkler, T. W. *et al.* Genome-wide association meta-analysis for early age-related macular degeneration highlights novel loci and insights for advanced disease. *BMC Med Genomics* **13**, 120 (2020).
11. Black, J. R. M. & Clark, S. J. Age-related macular degeneration: genome-wide association studies to translation. *Genetics in Medicine* **18**, 283–289 (2016).
12. Tedja, M. S. *et al.* Genome-wide association meta-analysis highlights light-induced signaling as a driver for refractive error. *Nat Genet* **50**, 834–848 (2018).
13. Hamel, A. R. *et al.* Integrating genetic regulation and single-cell expression with GWAS prioritizes causal genes and cell types for glaucoma. *medRxiv* 2022.05.14.22275022 (2022) doi:10.1101/2022.05.14.22275022.

14. ENCODE Project Consortium *et al.* Expanded encyclopaedias of DNA elements in the human and mouse genomes. *Nature* **583**, 699–710 (2020).
15. Ratnapriya, R. *et al.* Retinal transcriptome and eQTL analyses identify genes associated with age-related macular degeneration. *Nat Genet* **51**, 606–610 (2019).
16. Friedman, R. Z. *et al.* Information content differentiates enhancers from silencers in mouse photoreceptors. *Elife* **10**, (2021).
17. Cherry, T. J. *et al.* Mapping the cis-regulatory architecture of the human retina reveals noncoding genetic variation in disease. *Proc Natl Acad Sci U S A* **117**, 9001–9012 (2020).
18. BRINGMANN, A. *et al.* Müller cells in the healthy and diseased retina. *Prog Retin Eye Res* **25**, 397–424 (2006).
19. Zhang, X. *et al.* Functions of retinal astrocytes and Müller cells in mammalian myopia. *BMC Ophthalmol* **22**, 451 (2022).

Second round of review

Reviewer 1

The authors provided a comprehensive response and performed a substantial amount of new analyses and experiments to support their findings. Most of my comments have been successfully addressed. I have a few remaining suggestions:

1. The authors discovered that OTX2 binding was significantly enriched in experimentally validated enhancers. To support their model of hierarchical TF regulatory networks, the authors should consider referencing OTX2 as a pioneer factor.
2. The authors provided important new analysis to justify the potential gain in power for fine-mapping GWAS variants using single-cell multi-omic data. It would be interesting to include a comparison of single-cell RNA-seq over bulk RNA-seq, and single-cell multiome over single-cell RNA-seq, to give readers more contexts of the significance of their findings. In general, reporting these numbers in the main text or supplement would be helpful.
3. The authors acknowledged that small sample size is a crucial limiting factor in their analysis of cell type-specific regulatory networks. This does weaken the claim that “integration of single cell multiomics and GWAS studies can increase the resolution to nominate effective cell context”. I wonder if the authors have tried methods specifically designed to model correlation of effects in different tissue contexts, such as mash, to evaluate whether they can detect more cell type-specific effects.
4. Another minor comment related to cell type-specific regulatory effects is for eQTL (Fig 2C) and caQTL (Fig 4C) effect heatmaps, the diagonal can be grayed out to highlight comparisons between cell types. It seems that caQTL effects show more cell type specificity compared to eQTLs, which could be interesting to explore as well.

Overall, the authors made significant improvements of their manuscript and this work could be of great interest to the broad readership of Genome Biology, particularly if they could provide more insights on understanding of cell type-specific regulatory networks and disease associations.

Reviewer reports:

Reviewer #1: Please see above for full comments - the authors have successfully addressed most of my comments and significantly improved the manuscript.

Reviewer's Responses to Questions

Please provide your comments for the author(s) on the revised manuscript

Reviewer #1: The authors provided a comprehensive response and performed a substantial amount of new analyses and experiments to support their findings. Most of my comments have been successfully addressed. I have a few remaining suggestions:

1. The authors discovered that OTX2 binding was significantly enriched in experimentally validated enhancers. To support their model of hierarchical TF regulatory networks, the authors should consider referencing OTX2 as a pioneer factor.

Response:

Thanks Reviewer #1 for the insightful comment. We have revised the corresponding text as follows: "Interestingly, the validated enhancers are enriched of the co-binding of an early lineage-determining factor, *OTX2*, with at least one cell type/context specific trans-factor, i.e. *CRX*, *NEF2D* and *NRL*, compared to the inactive elements (62.5% vs. 46.2%, one-sided Fisher's exact test, $p = 6.2 \times 10^{-4}$, Fig. 7e). This result indicates the potential role of *OTX2* as a pioneer factor. "

2. The authors provided important new analysis to justify the potential gain in power for fine-mapping GWAS variants using single-cell multi-omic data. It would be interesting to include a comparison of single-cell RNA-seq over bulk RNA-seq, and single-cell multiome over single-cell RNA-seq, to give readers more contexts of the significance of their findings. In general, reporting these numbers in the main text or supplement would be helpful.

Response:

Thanks Reviewer #1 for the constructive comment. We have incorporated the analysis comparing single-cell RNA-seq to bulk RNA-seq into a section entitled "A comparison of fine-mapping using single-cell multi-omics data with existing data from bulk eQTL and annotated cis-regulatory regions" in Additional file 1: Supplementary Note.

Currently, conducting QTL analyses on large-scale single-cell multiome profiling data and comparing single-cell RNA-seq with single-cell multiome data is not feasible for us. However, we anticipate that single-cell multiome sequencing could further validate our findings and provide

novel insights. We have included this as a future direction in the last paragraph of the discussion section: "Another potential improvement is to simultaneously profile chromatin accessibility and gene expression within the same cell using single-cell multiome sequencing. This approach would allow for matching chromatin accessibility and allelic expression within individual cells, thereby increasing the power to validate our findings and gain further insights into gene regulation."

3. The authors acknowledged that small sample size is a crucial limiting factor in their analysis of cell type-specific regulatory networks. This does weaken the claim that "integration of single cell multiomics and GWAS studies can increase the resolution to nominate effective cell context". I wonder if the authors have tried methods specifically designed to model correlation of effects in different tissue contexts, such as mash, to evaluate whether they can detect more cell type-specific effects.

Response:

Thanks Reviewer #1 for the thoughtful suggestion. In our preliminary analysis, we explored methods to model the correlation of effects in different cell type contexts. Specifically, we utilized MetaTissue to identify eQTLs from multiple cell types and employed MetaSoft to perform meta-analysis and combine results from various cell types. This method, based on m-value per tissue, can predict whether a genetic variant exerts eQTL effect in a specific cell type^{1,2}. Consequently, we identified more eQTLs shared by multiple cell types compared to our current result. This result could also be partially attributed to the different FDR calculation approaches.

We have not explored the "mash" method. To address this comment, we have added the following text in the discussion section: "Increasing the sample size and refining the mapping methods, such as applying a linear mixed effect model or modeling the correlation of effects in different cell type contexts, would significantly enhance the power to detect QTLs."

4. Another minor comment related to cell type-specific regulatory effects is for eQTL (Fig 2C) and caQTL (Fig 4C) effect heatmaps, the diagonal can be grayed out to highlight comparisons between cell types. It seems that caQTL effects show more cell type specificity compared to eQTLs, which could be interesting to explore as well.

Response:

Thanks Reviewer #1 for the insightful suggestion. We modified Fig 2C and Fig 4C to gray out the diagonal. Indeed, it seems caQTL effects show more cell type specificity than eQTLs. We added the following text in the "Significant proportion of sc-caQTLs are cell type-specific in retina" section of the result: "Compared to the correlation of sc-eQTL effect sizes across cell types, those of sc-caQTLs appears to be smaller. This suggests that sc-caQTL effects show higher cell type-specificity than sc-eQTLs, possibly due to the greater variation in chromatin accessibility across cell types compared to gene expression."

Figure 2C: eQTL effect

Figure 4C: caQTL effect

Overall, the authors made significant improvements of their manuscript and this work could be of great interest to the broad readership of *Genome Biology*, particularly if they could provide more insights on understanding of cell type-specific regulatory networks and disease associations.

Response:

Thanks Reviewer #1 for the constructive comment. To provide insights on understanding of cell type-specific regulatory networks and disease associations, we added the following text to the discussion section: “The specificity of genetic variant effects depends on the context in which the trans-factors with motifs perturbed by genetic variants operate. Specifically, if genetic variants disrupt the binding of pioneer factors, the associated cis-elements could become inaccessible, preventing the binding of additional cofactors. Consequently, diseases may onset in broader cell types or spatial/temporal contexts. In contrast, if genetic variants disrupt the binding of late or context-dependent trans-factors, the cis-elements could become accessible by pioneer factors but hinder the binding of these late or context-dependent trans-factors. Consequently, diseases may be triggered in specific spatial and temporal cellular contexts. Additionally, since the same TFs and cis-elements can be utilized in different cellular contexts, diseases may occur in multiple cellular contexts. In summary, disease onsets could be determined by the spatial and temporal cellular contexts where the disrupted TF motifs are actively involved. These insights could facilitate the understanding of pathogenic mechanisms and development of treatments for diseases.”